# VLSA: Enhancing Vision-Language Understanding via Perception and Cognition Alignment

## Abstract

Prevalent Vision-Language (VL) alignment techniques within Multi-modal Large Language Models (MLLMs) struggle to adequately align the language model with visual inputs, resulting in hallucinations and undermining reliability. We rethink the modality alignment in MLLMs from the perspective of reducing information loss and present an efficient plug-in, VL Superior Alignment (VLSA), which decouples the alignment into two stages. The first stage, referred to as **Perception Alignment**, minimizes information loss in visual encoding through compressive encoding for high-resolution images and innovative reconstructive training leveraging latent diffusion models. The second stage, termed **Cognition Alignment**, reduces information loss in response generation by enhancing the language model's ability to grasp both high-level visual semantics and low-level image appearances, achieved by novel auxiliary self-supervised fine-tuning (SSFT) objectives. Extensive experiments across over 25 MLLM benchmarks and 7 MLLM architectures, thorough ablations, and analyses of computational overhead underscore the improvement of both performance and efficiency brought by VLSA. In service to the MLLM research community, our code and model checkpoints will be publicly available.

## 1 Introduction

Large language models (LLMs) are advanced tools for processing, understanding, and generating contextual information. In order to transform LLMs into powerful multi-modal LLMs (MLLMs), current techniques Bai et al. (2023); Li et al. (2023a); Gao et al. (2023); Liu et al. (2023a) generally adhere to standard processes involving using a pre-trained image encoder to embed visual context, then integrating the visual and textual embeddings into LLMs for a range of tasks.

The feasibility of this paradigm hinges on the premise that LLMs can process visual embeddings similarly to how they handle textual embeddings. Existing approaches like LLaVA Liu et al. (2023a), PaLI Chen et al. (2022), and CogVLM Wang et al. (2024a) employ a linear layer or MLP as the projector to bridge the gap among embeddings, while models such as BLIP2 Li et al. (2023a), InstructBLIP Dai et al. (2023), and Qwen-VL Bai et al. (2023) leverage Q-former (also named perceiver), which transfers arbitrary visual sequences into a fixed-length query through cross-attention, to achieve a similar effect. However, these Vision-Language (VL) alignment techniques primarily concentrate on achieving distributional or semantic consistency between VL embeddings, often overlooking the information loss during visual inference. Consequently, they may fail to adequately align the language model with visual inputs, leading to hallucinations and diminished reliability.

When contrasted to the ideal visual inference process illustrated at the top of Fig. 1(Left), which involves optimal visual encoding and response generation, actual visual inference empowered by conventional alignments shown at the bottom of Fig. 1(Left) highlights three distinct categories of information degradation that undermine system performance: (1) Original Information Loss (O-IL) stems from the limitations imposed by input resolution, where spatial downsampling of raw images inevitably discards critical visual details. (2) Encoding Information Loss (E-IL) arises from inherent limitations in current visual encoding architectures: both vision encoders and alignment projection modules function as lossy translators or compressors, compromising the information perceived by LLMs. (3) Decoding Information Loss (D-IL) occurs during the response generation, where LLMs demonstrate incomplete comprehension of encoded visual embeddings, causing the recognized visual information to diverge from what they perceived. Ultimately, D-IL leads to unreliable responses, even in scenarios with theoretically optimal visual encoding (i.e., when E-IL is absent).

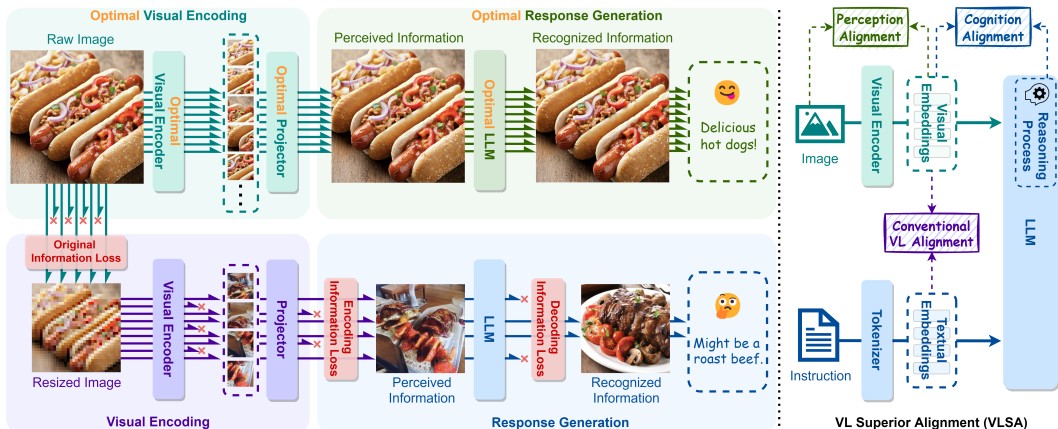

Figure 1: **Left:** Conventional Vision-Language (VL) alignments in MLLMs overlook information loss (IL) during visual inference. **Right:** To address this issue and further promote the alignment between the LLM and visual inputs, our VL Superior Alignment (VLSA) extends modality alignment in MLLMs beyond conventional VL alignments by incorporating Perception and Cognition Alignment.

Recent efforts Liu et al. (2024a); Li et al. (2023b); Bai et al. (2023); Zhang et al. (2023a); Young et al. (2024) have somewhat mitigated the O-IL by preserving or increasing the resolution of input images rather than uniformly resizing them to a smaller size during encoding. Typically, they involve segmenting a high-resolution image into patches, encoding each patch independently, and then concatenating them into a lengthy sequence that serves as visual embeddings. However, the efficiency of projector-based alignments (e.g., LLaVA) is diminished in this scenario, as the time complexity of LLMs scales quadratically with the input sequence length. Moreover, they rely on the LLM to interpret relationships among image patches. Nevertheless, the causal attention inherent in LLMs has limitations in accurately modeling these interrelationships, which can exacerbate the D-IL. (Notably, recent studies Xie et al. (2024a); Zhou et al. (2024) demonstrate remarkable improvements by integrating bi-directional attention for visual tokens within LLMs. Nonetheless, this modification alters the dynamics of LLM, leading to an increased demand for training data.) On the other hand, Q-former-based alignments (e.g., BLIP2) are lossy compressors that may overlook crucial features, intensifying the E-IL despite their potential to reduce visual sequence length for improved efficiency.

Moreover, most current techniques rely on supervised fine-tuning (SFT) tasks to promote LLMs' understanding of visual embeddings. Nevertheless, commonly used tasks, including image captioning, visual question answering, and optical character recognition, only help LLMs grasp certain aspects of visual semantics, potentially aggravating the D-IL. Recent progresses Zhang et al. (2023b); Wu et al. (2024); Li et al. (2024a); Lai et al. (2024) have broadened LLMs' comprehension by introducing additional training objectives, such as visual grounding and segmentation, which can alleviate the D-IL to some extent. Nonetheless, many of these supplementary tasks rely on manually annotated data or semi-automatic annotation processes that involve human participation, and they still tend to focus on particular attributes of visual semantics (Appendix B.2 provides ablations on these objectives).

To address these limitations, we broaden the goal of modality alignment in MLLMs to include minimizing information loss during visual inference, rather than focusing exclusively on aligning visual and textual embeddings. Additionally, as shown in Fig. 1(Right), we introduce the VL Superior Alignment (VLSA), which achieves modality alignment through two distinct stages: (1) **Perception Alignment**: This stage aligns visual embeddings with visual inputs, striving to minimize both O-IL and E-IL, all while reducing computational overhead. (2) **Cognition Alignment**: This stage aligns the LLM's reasoning process with visual embeddings, aiming to reduce D-IL. As depicted in Fig. 2, perception alignment involves compressive encoding for high-resolution images and a novel reconstructive training method inspired by the denoising process of Latent Diffusion Models (LDM) Esser et al. (2024). The former condenses the information from high-resolution images into their downsampled counterparts, utilizing a cross-attention-based module called SA-perceiver, which significantly reduces the sequence length of visual embeddings while effectively preserving

the spatial structure among image patches. The latter requires the model to reconstruct inputs from random noise (with outputs of compressive encoding as cues), enabling the image encoder to capture more details and allowing the SA-perceiver to function like a lossless compressor as much as possible. Concurrently, cognition alignment includes novel self-supervised fine-tuning (SSFT) objectives that augment the standard SFT objective, simultaneously facilitating the LLM's comprehensive understanding of high-level and low-level visual semantics. These auxiliary objectives leverage the codebook indices from a frozen VQ-VAE van den Oord et al. (2018) as discrete labels representing high-level semantics of images, while employing pixel values to convey low-level semantics. The MLLM is tasked with simultaneously predicting the responses to instructions, the codebook indices, and the pixel values of the images. To summarize:

- We rethink the modality alignment from the perspective of reducing information loss and propose an efficient plug-in VLSA, which decouples the alignment between the language model and visual inputs into two stages, including perception and cognition alignment.
- We present compressive encoding paired with LDM-based reconstructive training to prevent information loss in visual encoding and self-supervised fine-tuning objectives to reduce information loss in response generation.
- Comprehensive experimental evaluations across over 25 benchmarks and 7 MLLM architectures, along with rigorously constructed ablation studies, underscore the efficacy and essentiality of designs in VLSA.

## 2 RELATED WORK

Some recent refinements of MLLMs can be interpreted through the lens of reducing information loss.

**Multi-resolution visual features.** Works such as Monkey Li et al. (2023c), LLaVA-UHD Xu et al. (2024), and SliME Xu et al. (2024) leverage both low and high-res visual features to improve performance, effectively reducing O-IL with reasonable computational efficiency. However, Monkey and LLaVA-UHD employ Q-former-like resamplers to reduce visual tokens, which, despite improving efficiency, may exacerbate E-IL, resulting in slower convergence and suboptimal performance (Fig. 6, Tab. 9(ex7) in Appendix). Similar to our approach, SliME uses low-resolution images as reasoning foundations and high-resolution patches as supplements. However, SliME implements a complex MOE adapter with a compression layer and a prompt-based router for crucial high-res tokens, requiring three-stage alternating training and additional data. In contrast, our VLSA comprehensively mitigates information loss while enhancing efficiency when applying multi-resolution inputs.

**Multiple visual encoders.** Approaches like BRAVE Kar et al. (2024), COMM Jiang et al. (2024), and LLaVA-Read Zhang et al. (2024a) leverage multiple visual experts to reduce E-IL. Despite performance gains, this introduces significant computational overhead. Our VLSA simultaneously utilizes CLIP and VQ-VAE encoders, with VQ-VAE serving exclusively as an annotation tool to generate training data for SSFT objectives within Cognition Alignment. The entire VQ-VAE annotation process can occur offline, preserving computational efficiency.

**Reconstructive training objectives.** Advances like X-former Swetha et al. (2024), LaViT Jin et al. (2024), and Ross Wang et al. (2024b) explore reconstructive training to reduce information loss. X-former employs MAE to recover local details missing from CLIP's global features, addressing encoder-related E-IL but leaving projector-induced E-IL. LaViT reconstructs outputs from its frozen visual encoder (not raw images), minimizing its vector quantization loss while failing to address encoder-related E-IL. Ross uses LLM-processed image tokens as reconstruction clues, effectively reducing E-IL but interfering with the LLM's training and leading to suboptimal performance (please refer to Appendix B.10 for more analysis). Conversely, our VLSA comprehensively reduces information loss during visual perception without compromising LLM training, while also promoting semantic alignment between VL features.

**Vision-Language early fusion.** Methods such as QA-ViT Ganz et al. (2024), EMMA Ghazanfari et al. (2025), and mPLUG-Owl2 Ye et al. (2023) enhance performance by modifying Visual-Language (VL) feature fusion locations, enabling preliminary VL interactions before visual features enter the LLM. Compared to standard approaches, early fusion mitigates loss of critical visual information by filtering visual semantics to focus on user requests, boosting performance on standard VQA tasks. However, this approach presents a trade-off: performance degrades with ambiguous or overly verbose user queries (see Appendix B.9 for more analysis).

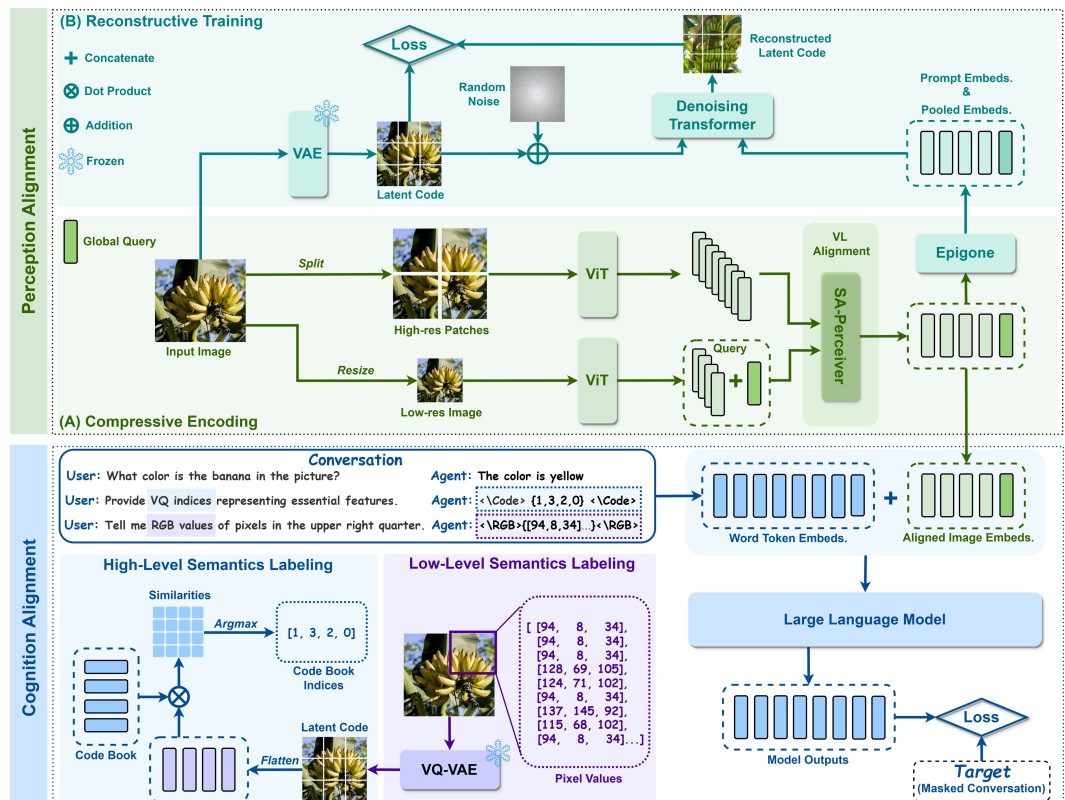

Figure 2: The Architecture of VLSA. TOP: Perception Alignment consists of (a) compressive encoding relieving O-IL while reducing system overhead, and (b) reconstructive training minimizing E-IL. Bottom: Cognition Alignment utilizes auxiliary training objectives to reduce D-IL.

## 3 METHODOLOGY

### 3.1 PERCEPTION ALIGNMENT

**Compressive encoding for high-resolution images.** VLSA takes an image $X$ with arbitrary height and width as input. The initial step involves padding $X$'s dimension to $\mathbb{R}^{3 \times H \times W}$, which is the smallest dimension that can be evenly divided by $(h, w)$, the input size of the image encoder. Following this, $X$ is split into high-resolution patches $X_{\mathrm{Hi}} \in \mathbb{R}^{m \times 3 \times h \times w}$, with $m$ representing the number of patches. Simultaneously, $X$ is downsampled to create a low-resolution snapshot $X_{\mathrm{Lo}} \in \mathbb{R}^{3 \times h \times w}$. Both $X_{\mathrm{Hi}}$ and $X_{\mathrm{Lo}}$ are then separately encoded by the CLIP Radford et al. (2021) vision encoder, resulting in visual embedding $V_{\mathrm{Hi}} \in \mathbb{R}^{m \times l \times d}$ and $V_{\mathrm{Lo}} \in \mathbb{R}^{l \times d}$, where $l$ is the sequence length and $d$ is the feature dimension. Differing from previous techniques that flatten multi-scale visual embeddings to a single sequence, we gather information from $V_{\mathrm{Hi}}$ while preserving their spatial structure according to $V_{\mathrm{Lo}}$. This goal is achieved through the SA-Perceiver depicted in Fig. 3. Specifically, we append a learnable query $q \in \mathbb{R}^{1 \times d}$ to $V_{\mathrm{Lo}}$ to gather global visual features, resulting $V'_{\mathrm{Lo}} \in \mathbb{R}^{(l+1) \times d}$. Subsequently, we update $V'_{\mathrm{Lo}}$ with $V_{\mathrm{Hi}}$ by cross-attention:

$$Q = w_{\mathrm{q}} V'_{\mathrm{Lo}}, \quad K = w_{\mathrm{kv}} V_{\mathrm{Hi}}, \quad V = w_{\mathrm{kv}} V_{\mathrm{Hi}},$$
$$V'_{\mathrm{Lo}} = \sigma \left( w_{\mathrm{o1}} \left( Q \times K \right) V \right) + V'_{\mathrm{Lo}}, \tag{1}$$

where $w_{\mathrm{q}}, w_{\mathrm{kv}}, w_{\mathrm{o1}}$ with the dimension of $\mathbb{R}^{d \times d}$ are linear projectors, $\sigma$ is the activation function SiLU Elfwing et al. (2018). After that, we utilize a self-attention layer on $V'_{\mathrm{Lo}}$ to further process the aggregated information from $V_{\mathrm{Hi}}$ and facilitate the global feature extraction:

$$V'_{\mathrm{Lo}} = w_{\mathrm{o2}} \left( \left( V'_{\mathrm{Lo}} \times w_{\mathrm{k}} V'_{\mathrm{Lo}} \right) \right) V'_{\mathrm{Lo}}. \tag{2}$$

Finally, we split $V'_{\mathrm{Lo}}$ into visual embeddings $V \in \mathbb{R}^{l \times d}$ and global embedding $P \in \mathbb{R}^{d}$ for the subsequent reconstructive training. Furthermore, the SA-Perceiver is also responsible for achieving VL alignment, similar to the role of the MLP projector in LLaVA or the Q-former in BLIP-2.

Compared with previous routines, our method mitigates the challenge of modeling visual content solely relying on causal attention within LLMs. Also, it lowers the computational overhead in the subsequent LLMs' reasoning by reducing the sequence length of visual embeddings.

**Reconstructive training with LDM.** To minimize information loss during image encoding, we ensure that visual embeddings can accurately reconstruct the input image $X$. Drawing inspiration from Latent Diffusion Models (LDM), we initiate this reconstruction from Gaussian noise, with $V$ and $P$ serving as vital clues. As illustrated in Fig. 2 (B), the process begins with encoding $X$ into the latent code $z \in \mathbb{R}^{s \times d}$ through a frozen Variational Auto-Encoder (VAE) Kingma (2013), where $s$ and $d$ represent the length and dimension, respectively. $z$ will be served as the target for reconstructive training. Following this, we add Gaussian noise onto $z$, the intensity of which depends on the time step $t$, yielding a noisy latent $z_t$. The crux of reconstructive training lies in optimizing a denoising transformer that reconstructs $z$ from $z_t$ using $V$ and $P$ as guidance. Both VAE and the denoising transformer are initialized from the text-to-image LDM, Stable Diffusion 3-medium (SD) Esser et al. (2024). The reconstruction loss is formulated as:

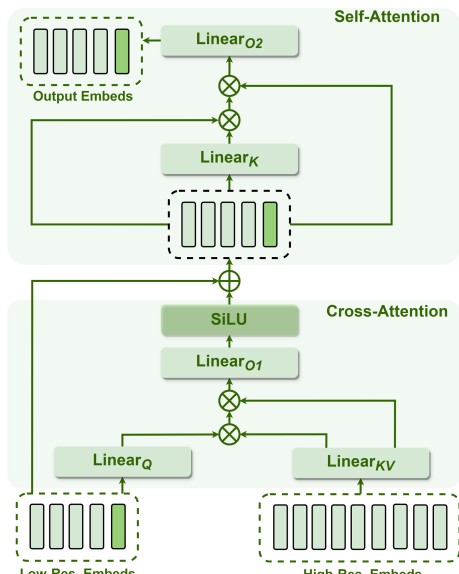

$$\mathcal{L}_{\text{rec}} = \| z - z_\theta(z_t, \, c) \|_2^2, \qquad (3)$$

Figure 3: The details of the SA-Perceiver.

where $z_\theta$ is the denoised latent predicted by the denoising transformer with learnable parameter $\theta$, $c$ represents the projected concatenation of prompt embeddings and pooled embeddings carrying linguistic semantics required by SD.

$$c = Epigone([V, \, P]). \qquad (4)$$

Here, the Epigone serves as a rough translator, converting visual embeddings into prompt embeddings. Unlike the word embedding layer in SD, which conveys natural language's concise high-level semantics, the Epigone is tailored to transmit detailed image information. As a result, it mitigates the randomness (detrimental to reconstruction) inherent in SD. To enhance computational efficiency, we adopt a simple MLP as the Epigone. However, it is conceivable that a more elaborate design could further boost performance.

*The primary advantage of utilizing Latent Diffusion Models (LDMs) over traditional Auto-Encoders (AEs) in reconstructive training lies in pretrained text-to-image LDMs not only effectively minimize information loss during image encoding, but also assist the SA-Perceiver in reaching semantic alignment between visual and textual embeddings.* This benefit arises from the superior capability of text-to-image LDMs to generate images based on linguistic semantics.

### 3.2 COGNITION ALIGNMENT

**Auxiliary self-supervised objectives.** To minimize information loss during response generation, we augment general supervised fine-tuning (SFT) objectives of visual instruction tuning with additional self-supervised fine-tuning (SSFT) objectives to strengthen the LLM in recognizing high-level and low-level visual semantics simultaneously. To discretely label high-level semantics in a self-supervised manner, we first calculate the similarity between the image $X$'s latent code $L \in \mathbb{R}^{l' \times d'}$, which is generated by a frozen VQ-VAE (Vector Quantized VAE) van den Oord et al. (2018) initialized from Tang et al. (2023), and VQ-VAE's codebook $B \in \mathbb{R}^{b \times d'}$. Here, $l'$ and $b$ are the lengths of the latent code and the codebook, $d'$ is the dimension of VQ-VAE's latent space. Then, we leverage the codebook indices of embeddings in $B$ that have the highest similarity with $L$ as the label (denote as $Target_{\text{VQ}} \in \mathbb{R}^{l'}$) for high-level semantics. After simplification, this process can be formulated as:

$$Target_{\text{VQ}} = \text{argmin}(B \times L). \qquad (5)$$

On the other hand, the labels of low-level semantics are simply represented by RGB values of pixels in $X_{\text{Lo}}$ from Sec 3.1, denoted as $Target_{\text{PX}} \in \mathbb{R}^{3 \times h \times w}$. Our SSFT objectives require the LLM to

Table 1: **Comparisons on academic-task-oriented datasets**. *denotes a larger actual receptive field. †Includes in-house data that is not publicly accessible. Res, PT, and IT indicate image resolution and the number of samples in pretraining and finetuning, respectively.

| Method | LLM | Res. | PT | IT | VQA$^{v2}$ | GQA | VisWiz | COCO | TextC | VQA$^{ST}$ | SQA$^I$ | SQA | VQA$^T$ |
|---|---|---|---|---|---|---|---|---|---|---|---|---|---|
| BLIP-2 | Vicuna-13B | 224 | 129M | - | 41.0 | 41 | 19.6 | – | – | 36.4 | 61 | - | 42.5 |
| InstructBLIP | Vicuna-13B | 224 | 129M | 1.2M | – | 49.5 | 33.4 | – | – | 38.7 | 63.1 | - | 50.7 |
| Shikra | Vicuna-13B | 224 | 600K | 5.5M | 77.4 | – | – | – | – | – | – | - | – |
| IDEFICS-80B | LLaMA-65B | 224 | 353M | 1M | 60.0 | 45.2 | 36.0 | – | 56.8 | – | – | - | 30.9 |
| Qwen-VL-Chat | Qwen-7B | 448 | 1.4B† | 50M† | 78.2 | 57.5 | 38.9 | – | – | – | 68.2 | - | **61.5** |
| LLaVA-1.5 | Vicuna-7B | 336 | 558K | 665K | 76.6 | 62.0 | 50.0 | 109.4 | 101.8 | 54.0 | 69.8 | 70.0 | 58.2 |
| LLaVA-1.5-HD | Vicuna-7B | Any | 558K | 665K | 81.2 | 64.6 | 54.1 | 125.4 | 107.3 | 63.9 | 71.2 | 71.8 | 62.1 |
| +VLSA | Vicuna-7B | 336* | 558K | 665K | 83.0 | 65.2 | **61.5** | 129.3 | **107.5** | 65.0 | 73.9 | 76.5 | 63.0 |
| Δ | - | - | - | - | +1.8 | +0.6 | +7.4 | +3.9 | +0.2 | +1.1 | +2.7 | +4.7 | +0.9 |
| LLaVA-Next | LLaMA3-8B | Any | 558K | 790K | 82.4 | 64.9 | 46.7 | 137.3 | 70.1 | 64.2 | 74.6 | 72.1 | 63.9 |
| +VLSA | LLaMA3-8B | 336* | 558K | 790K | **83.5** | 65.3 | 57.7 | **139.5** | 73.3 | **65.7** | **77.5** | **78.6** | **65.2** |
| Δ | - | - | - | - | +1.1 | +0.4 | +11.0 | +2.2 | +3.2 | +1.5 | +2.9 | +6.5 | +1.3 |

predict $Target_{VQ}$ and $Target_{PX}$ from visual embeddings $V$ in Sec 3.1. In practice, we modify $Target_{PX}$ to be pixel values within a randomly selected quarter of $X_{Lo}$ to reduce the overhead.

*Unlike previous attempts to enhance the understanding of visual semantics in specific categories, our approach offers a more holistic visual comprehension without manual annotation.* This is because the VQ-VAE we employed functions as an autoencoder, with its codebook indices capable of encapsulating a broad spectrum of semantics that transcend specific, human-defined categories.

*Our intuition behind the effectiveness of predicting codebook indices:* During SFT, the LLM partially understands the mapping between visual semantics and natural language. Meanwhile, it grasps the majority of the mappings between visual semantics and codebook indices during our SSFT. Given the overlap in the visual semantics present in both mapping sets, the LLM is able to establish connections between natural language and codebook indices. Consequently, SSFT indirectly enhances the LLM's ability to map visual semantics to natural language. Please refer to Appendix B.5 for more analysis.

**Joint training with multiple objectives.** We incorporate instructions for predicting codebook indices and pixel values (please refer to templates provided in Appendix B.11 ) alongside their corresponding answers $Target_{VQ}$ and $Target_{PX}$ into SFT datasets for visual instruction tuning (e.g., LLaVA-665k Liu et al. (2023b)). By randomly selecting positions within conversations to insert these instruction-answer pairs, we enable our SSFT objectives to assist the original SFT objective and further improve visual reasoning. For each instruction in the augmented dataset, we encode it into word token embeddings $I_{emb}$, which are then concatenated with image embeddings $V$, serving as inputs for the LLM to generate the response. We adopt the same language modeling loss $\mathcal{L}$ as LLaVA Liu et al. (2023a) (see Appendix B.6). Finally, the overall optimization loss for the entire system is formulated as $\mathcal{L}_{all} = \mathcal{L} + \mathcal{L}_{rec}$.

# 4 EXPERIMENTS

To assess the capabilities of VLSA, we integrate it into prevalent MLLM architectures MiniGemini Li et al. (2024b), Qwen2.5-VL Bai et al. (2025), InternVL-2.5 Chen et al. (2025a), LLaVA-UHD Xu et al. (2024), SliME Zhang et al. (2024b), LLaVA-1.5-HD Liu et al. (2023b), and LLaVA-Next Dubey et al. (2024).

The discussions in this chapter primarily focus on comparing VLSA with the LLaVA series, which employs a simple MLP projector alongside basic SFT training to achieve modal alignment and demonstrate strong performance. This conciseness allows for a better analysis of the impact of the components of VLSA. Unless otherwise specified, performances of LLaVA-1.5-HD and LLaVA-Next are derived from variants that utilize CLIP-ViT-L/14@336px as the visual encoder. We maintain the two training stages when integrating VLSA with the LLaVA series. In **Stage I**, we prioritize optimizing the SA-perceiver, the Epigone, and the denoising transformer, while keeping other parameters fixed. The dataset used in this stage matches the 558K instances previously employed in pre-training LLaVA-1.5-HD and LLaVA-Next. Transitioning to **Stage II**, comprehensively optimize all model parameters, utilizing the fine-tuning dataset containing 665K/790K instructions as leveraged by LLaVA-1.5-HD/LLaVA-Next. Please refer to Appendix B.1 for more implementation details.

Table 2: **Comparison for multi-modal comprehension on MLLM benchmarks**. ‡Evaluating via text-only GPT-4-0613. *denotes a larger actual receptive field.

| Method | LLM | Res. | PT | IT | POPE | MME | MMB | MMB$^{CN}$ | SEED | LLaVA$^W$ | MM-Vet |
|--------|-----|------|-----|-----|------|-----|-----|------|------|------|--------|
| BLIP-2 | Vicuna-13B | 224 | 129M | - | 85.3 | 1293.8 | – | – | 46.4 | 38.1‡ | 22.4‡ |
| InstructBLIP | Vicuna-13B | 224 | 129M | 1.2M | 78.9 | 1212.8 | – | – | – | 58.2‡ | 25.6‡ |
| Shikra | Vicuna-13B | 224 | 600K | 5.5M | – | – | 58.8 | – | – | – | – |
| IDEFICS-80B | LLaMA-65B | 224 | 353M | 1M | – | – | 54.5 | 38.1 | – | – | – |
| Qwen-VL-Chat | Qwen-7B | 448 | 1.4B$^†$ | 50M$^†$ | – | 1487.5 | 60.6 | 56.7 | 58.2 | – | – |
| LLaVA-1.5 | Vicuna-7B | 336 | 558K | 665K | 85.9 | 1462.6 | 64.8 | 57.6 | 58.6 | 63.2‡ | 30.6‡ |
| LLaVA-1.5-HD | Vicuna-7B | Any | 558K | 665K | 87.1 | 1505 | 67.5 | 64.0 | 61.3 | 69.5‡ | 31.2‡ |
| +VLSA | Vicuna-7B | 336* | 558K | 665K | 87.8 | 1622 | 68.3 | 64.8 | **67.1** | 70.5‡ | 35.9 |
| Δ | - | - | - | - | +0.7 | +117 | +0.8 | +0.8 | +5.8 | +1.0 | +4.7 |
| LLaVA-Next | LLaMA3-8B | Any | 558K | 790K | 87.7 | 1753 | **72.2** | 70.1 | 60.0 | 69.4‡ | 33.4‡ |
| +VLSA | LLaMA3-8B | 336* | 558K | 790K | **88.6** | **1844** | 71.9 | **71.5** | 65.6 | **72.1‡** | **39.7‡** |
| Δ | - | - | - | - | +0.9 | +91 | -0.3 | +1.4 | +5.6 | +2.7 | +6.3 |

## 4.1 EXPERIMENTAL RESULTS

**Quantitative Results.** *To demonstrate the effectiveness of VLSA,* we first integrate it into the LLaVA series and compare the performances with BLIP-2 Li et al. (2023a), InstructBLIP Dai et al. (2023), Shikra Chen et al. (2023), IDEFICS Laurençon et al. (2023), and Qwen-VL Bai et al. (2023). Tab. 1 reports performances on 8 academic-task-oriented datasets (each examines a specific ability), including 6 VQA tasks: VQA$^{v2}$ Goyal et al. (2017), GQA Hudson and Manning (2019), VisWiz Gurari et al. (2018), ST-VQA Biten et al. (2019), ScienceQA-IMG Lu et al. (2022), TextVQA Singh et al. (2019) and 2 image captioning tasks: TextCaps Sidorov et al. (2020), COCO Chen et al. (2015). The table shows that VLSA demonstrates a significant and consistent improvement compared to the baselines. Specifically, VLSA notably enhances performance on unseen datasets VisWiz (+7.4/+11.0) and ScienceQA (+4.7/+6.5), which are not included in the training sets. The following are analyses of this phenomenon:

VizWiz's distinctive feature lies in the images captured by visually impaired users, which often exhibit heavy blurring, poor lighting, and incomplete content. Baselines may struggle to parse such degraded visual inputs, leading to hallucinations during testing. In contrast, VLSA, through Perception Alignment, better preserves valuable visual semantics from low-quality images during encoding, thereby enhancing the reliability of question answering. On the other hand, SQA's characteristic lies in questions often accompanied by lengthy lectures or hints to guide logical reasoning. However, verbose textual contexts might interfere with baseline models in interpreting and leveraging visual embeddings. VLSA alleviates this issue via Cognition Alignment, which mitigates the LLM's tendency to misinterpret or overlook semantics in visual embeddings within long-context scenarios.

However, VLSA does not lead to a notable performance boost on VQA$^{v2}$, GQA, and COCO. This observation may be attributed to the inclusion of images or annotations from these datasets during the fine-tuning process, utilizing 665K/790K data from LLaVA. Consequently, the baseline model also performs well on these datasets.

Besides, we further assess VLSA's multi-modal comprehension ability on 7 instruction-following benchmarks that are tailor-made for MLLMs, including POPE Li et al. (2023d), MME Fu et al. (2023), MMBench Liu et al. (2023c), MMBench-Chinese Liu et al. (2023c), SEED-Bench Li et al. (2023e), LLaVA-Bench (In-the-Wild) Liu et al. (2023a) and MM-Vet Yu et al. (2023). Tab. 2 shows that VLSA achieves obviously better results compared to previous generalist models on MME (+117/+91), SEED (+5.8/+5.6), and MM-Vet (+4.7/+6.3).

Table 3: **Generalizability of VLSA to more MLLM architectures.**

| Method | LLM | GQA | VQA$^T$ | MMB | MME | MM-Vet | LLaVA$^W$ | MMMU$_{val}$ |
|--------|-----|-----|------|-----|-----|--------|------|--------|
| MiniGemini-HD | Vicuna-7B | 66.4 | 68.4 | **65.8** | 1865 | 41.3 | 63.0 | 36.8 |
| +VLSA | Vicuna-7B | **66.9** | **69.5** | 65.6 | **1944** | **49.2** | **65.2** | **37.3** |
| Δ | - | +0.5 | +1.1 | -0.2 | +79 | +7.9 | +2.2 | +0.5 |
| Qwen2.5-VL | Qwen2.5-7B | 68.7 | 84.1 | 82.8 | 2303 | 62.5 | 74.7 | 58.6 |
| +VLSA | Qwen2.5-7B | **69.0** | **86.2** | **85.8** | **2354** | **66.3** | **74.8** | **61.4** |
| Δ | - | +0.3 | +2.1 | +3.0 | +51 | +3.8 | +0.1 | +2.8 |
| InternVL-2.5 | InternLM2.5-7B | 78.2 | 77.4 | 84.3 | 2322 | 64.9 | 78.5 | 53.8 |
| +VLSA | InternLM2.5-7B | 78.2 | **79.6** | **84.8** | **2358** | **66.1** | **79.4** | **56.1** |
| Δ | - | +0.0 | +2.2 | +0.5 | +36 | +1.2 | +0.9 | +2.3 |
| LLaVA-UHD | Vicuna-13B | 65.3 | 65.0 | 67.8 | 1535 | 33.8 | 64.2 | 37.1 |
| +VLSA | Vicuna-13B | **65.7** | **66.1** | **68.2** | **1573** | **35.6** | **65.7** | **37.4** |
| Δ | - | +0.4 | +1.1 | +0.4 | +38 | +1.8 | +1.5 | +0.3 |
| SliME | LLaMA3-8B | 64.6 | 64.1 | 70.9 | 1861 | 35.4 | 70.2 | 42.6 |
| +VLSA | LLaMA3-8B | **65.3** | **64.9** | **74.6** | **1887** | **38.0** | **70.8** | 46.0 |
| Δ | - | +0.7 | +0.8 | +3.7 | +26 | +2.6 | +0.6 | +3.4 |

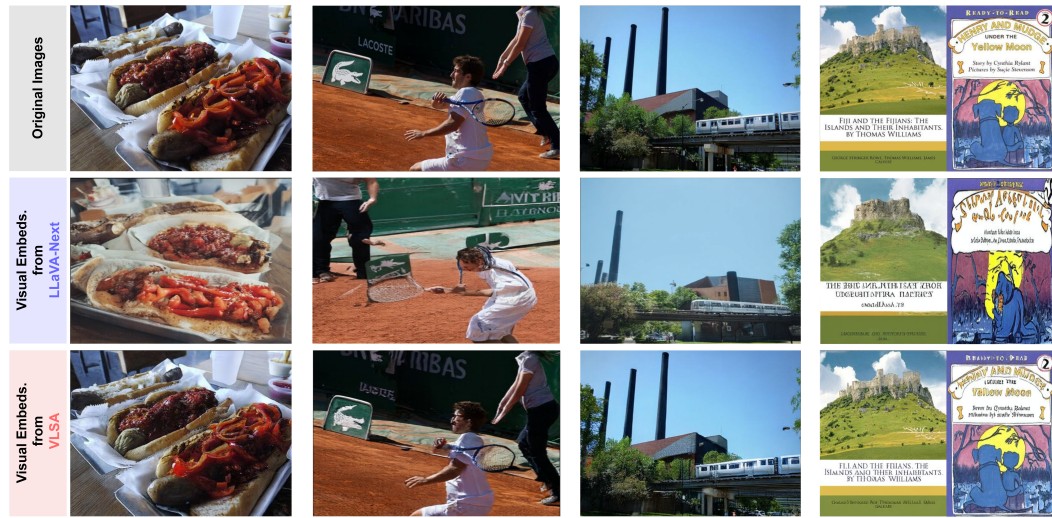

Figure 4: Perception alignment significantly reduces information loss during visual encoding.

In addition, we compare VLSA with LLaVA-Next on document understanding tasks AI2D Kembhavi et al. (2016), ChartQA Masry et al. (2022), and DocVQA Mathew et al. (2021) in Tab. 4. Upon comparing Tab. 4 (6) and (1), we find that despite adopting the compressive encoding on visual inputs, VLSA still achieves better results than the typical high-resolution approach on tasks requiring fine-grained perceptual capabilities. Please also refer to Appendix B.7 for evaluations on the other six carefully selected benchmarks.

*To demonstrate the generalizability of VLSA,* we further integrate VLSA with other MLLM architectures. As shown in Tab. 3, VLSA brings stable improvements to these architectures. Note that we initialized the models using the officially released weights (and without changing model architectures by integrating our SA-Perceiver) for experiments related to Qwen2.5 VL and Intern2.5 VL instead of training them from scratch, as we do not have access to their training data. Subsequently, we performed a secondary fine-tuning for them on LLaVA-Next's 790K fine-tuning set. Similarly, SliME is also involved in-house data, thus we reproduce the results using LLaVA-Next's training sets.

**Qualitative Results.** To underscore the efficacy of our perception alignment, we conduct an analytical visualization comparing reconstructed images based on visual embeddings from LLaVA-Next and our VLSA. As shown in Fig. 4, the baseline trends overlook substantial semantics in the original images. By contrast, our VLSA with reconstructive training minimizes the loss of details. To further demonstrate the advantages of our perception alignment and cognition alignment, we provide additional results on visual writing and visual question answering tasks in Appendix B.12.

**Efficiency Evaluation.** To evaluate VLSA's influence on computational efficiency, we randomly select 1,000 instances from the 790K pre-training data, strip away textual instructions, resize images to various resolutions, and compare the total latency and FLOPs of VLSA against the baseline when handling these inputs (operating on a single Nvidia A100 80GB GPU with a batch size of 1). As shown in Fig. 5, while VLSA introduces multi-scale visual feature interactions in SA-Perceiver and incorporates reconstructive training, it still significantly enhances computational efficiency compared to the baseline. Besides, the integration of VLSA also reduces the overall training hours of LLaVA-Next from 35.3 to 18.9 (-47%). Please refer to Appendix B.3 for more analysis on efficiency.

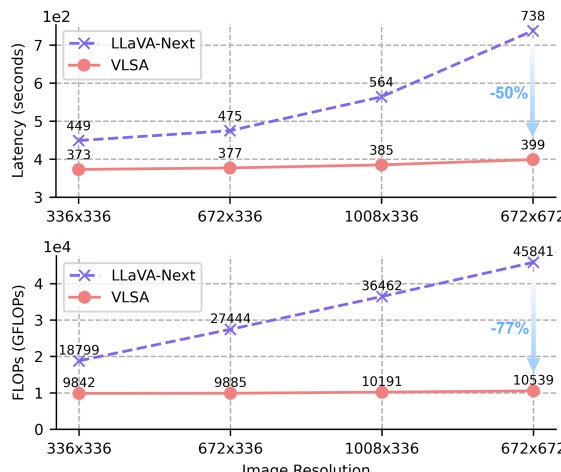

Figure 5: Latency and FLOPs in processing 1,000 images under various resolutions.

Table 4: Results on document understanding tasks and ablation studies.

| Method | Res. | AI2D | ChartQA | DocVQA | MME$^C$ | MME$^P$ | SQA | VisWiz |
|---|---|---|---|---|---|---|---|---|
| (1) LLaVA-Next | Any | 69.5 | 67.1 | 73.7 | 298.9 | 1455.7 | 72.1 | 46.7 |
| *Ablations on the Low Resolution Setting* | | | | | | | | |
| (2) LLaVA-Next | 336 | 67.8 (-1.7) | 44.8 (-22.3) | 45.4 (-28.3) | 322.1 (+23.2) | 1417.3 (-38.4) | 78.3 (+6.2) | 48.9 (+2.2) |
| *Ablations on the Perception Alignment* | | | | | | | | |
| (3) LLaVA-Next + (a) | 336* | 69.7 (+0.2) | 65.3 (-1.8) | 61.8 (-11.9) | 315.0 (+7.1) | 1426.5 (-29.2) | 78.1 (+6.0) | 55.9 (+9.2) |
| (4) LLaVA-Next + (ab) | 336* | 68.2 (-1.3) | 67.4 (+0.3) | **75.5** (+1.8) | 329.9 (+40.0) | 1481.4 (+25.7) | 74.1 (+2.0) | 53.6 (+6.9) |
| *Ablations on the Cognition Alignment* | | | | | | | | |
| (5) LLaVA-Next + (c) | Any | **72.5** (+3.0) | 67.0 (-0.1) | 73.9 (+0.2) | **353.6** (+54.7) | **1570.0** (+114.3) | 72.6 (+0.5) | 50.9 (+4.2) |
| (6) LLaVA-Next + (abc) | 336* | 71.4 (+1.9) | **67.9** (+0.8) | 75.2 (+1.5) | 336.4 (+37.5) | 1507.4 (+51.7) | **78.6** (+6.5) | **57.7** (+11.0) |

**More Evaluations.** Besides results on the above benchmarks, we provide two additional assessments on the efficiency of VLSA in facilitating modality alignment in Appendix B.4.

## 4.2 ABLATION STUDY

We delve into an exhaustive exploration of the impact of components within VLSA, including (a) compressive encoding, (b) reconstructive training, and (c) cognition alignment. To this end, we compare the results of six variants across eight benchmarks. Variants are (1) the baseline LLaVA-Next, (2) LLaVA-Next with the reduced resolution, (3) LLaVA-Next incorporating (a), (4) LLaVA-Next integrating (a) and (b), achieving our proposed perception alignment, (5) LLaVA-Next solely employing (c), and (6) full VLSA. The eight datasets, categorized by the abilities they assess, include ChartQA, DocVQA, and MME-perception, which primarily evaluate perceptive capabilities; MME-cognition, focusing on cognitive abilities; and AI2D, SQA, and VizWiz, which comprehensively assess both perception and cognition.

**Effectiveness of Perception Alignment.** Tab. 4 (1) and (2) indicate that reducing the input resolution severely impairs perceptual capabilities. However, (2) outperforms the baseline on MME-cognition, SQA, and VisWiz, revealing that previous techniques do not effectively model high-resolution inputs. Through (1), (2), and (3), we observe that (a) alleviates the perceptual challenges posed by low-resolution inputs while retaining the benefits of (2). Upon comparing (4) and (3), it becomes evident that (b) further improves the perceptual abilities to a point where it exceeds the baseline model by reducing the information loss in visual encoding.

**Effectiveness of Cognition Alignment.** By comparing (6) with (4), we find that (c) encourages the LLM to understand both shallow and deep semantics of visual embeddings, leading to consistent performance improvements. We also conducted isolated testing on (c) in (5),

Table 5: Ablations on SSFT objectives.

| Pixel Values | Codebook indices | AI2D | SQA | ChartQA |
|---|---|---|---|---|
| ✗ | ✗ | 68.2 | 74.1 | 67.4 |
| ✗ | ✓ | 69.8 (+1.6) | 77.2 (+3.1) | 67.7 (+0.3) |
| ✓ | ✗ | 68.4 (+0.2) | 71.6 (-2.5) | 67.5 (+0.1) |
| ✓ | ✓ | **71.4** (+3.2) | **78.6** (+4.5) | **67.9** (+0.5) |

which resulted in significant performance improvements on AI2D, MME-cognition, MME-perception, and VisWiz, highlighting the universal value of cognition alignment. In addition, we report ablations on the two SSFT objectives in Tab. 5.

**More Abations.** In Appendix B.2, we provide supplementary ablation studies that investigate the different factors affecting the performance of VLSA. These factors include (1) the structural design of the SA-Perceiver, (2) the LDM utilized in reconstruction training, (3) various image encoders, (4) different LLM backbones, (5) the ratio between language modeling loss and reconstruction loss, and (6) other tasks to facilitate the LLM's understanding of visual semantics.

## 5 DISCUSSION AND CONCLUSION

This paper investigates the modality alignment in MLLMs and introduces the concept of perception alignment, which aims to minimize information loss during visual encoding, and cognition alignment, which seeks to reduce information loss in response generation. Additionally, the paper presents VLSA, an efficient plug-in that employs compressive encoding and reconstructive training to achieve perception alignment while minimizing computational overhead. It also includes supplementary SSFT objectives that improve LLMs in comprehending both high-level and low-level image semantics, thereby facilitating cognition alignment. Extensive experiments validate the effectiveness of VLSA.

**Limitations & Future Work.** Given the multiple optimization objectives, the current approach simultaneously leverages them throughout all training phases, which may be suboptimal. It could be beneficial to explore reasonable strategies for their application during various training stages to enhance performance further.

## 5.1 REPRODUCIBILITY STATEMENT

We provide the source code and configuration for the main experiments, including scripts to generate data and train the models. All proofs are stated in the appendix, along with explanations and underlying assumptions. We thoroughly checked the implementation and also verified empirically that the proposed VLSA framework holds.

## 5.2 ETHICS STATEMENT

Improving the reliability and reducing hallucinations in Multi-modal Large Language Models (MLLMs) through techniques like VLSA has significant positive implications, enabling more trustworthy AI systems for diverse applications such as education, accessibility, and scientific analysis. However, like advances in generative AI, this work carries potential for misuse, such as creating more convincing synthetic content or deepfakes. We believe the substantial benefits of developing more reliable and aligned AI systems outweigh these potential negative applications.

Not addressing the fundamental issue of information loss and hallucination in MLLMs poses a greater risk, as unreliable models could cause harm through misinformation or poor decision-making in critical applications. By making our code and model checkpoints publicly available, we aim to foster transparency and responsible development within the research community. While we did not conduct dedicated experiments on fairness or bias mitigation in this work, we acknowledge that biases present in training data could persist and encourage future research to address these important concerns.

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

CONTENTS

## A    USE OF LARGE LANGUAGE MODELS (LLMS)

We declare the use of Large Language Models (LLMs) in this research work. The LLMs serve a supportive role in the following aspects of this project:

**Writing and Language Polishing:** LLMs assist in improving the clarity, readability, and grammatical correctness of the manuscript. This includes refining sentence structure, improving word choice, and ensuring consistent terminology throughout the paper.

**Literature Review Support:** LLMs assist in reading and summarizing research literature to identify relevant prior work and contextualize our contributions within the existing body of knowledge. This includes assistance with understanding complex technical concepts and identifying key papers in the field.

The core research ideas, experimental design, theoretical framework, and scientific contributions presented in this work are original contributions by the authors. The LLMs do not contribute to the fundamental research conception, hypothesis formulation, or interpretation of results. All experimental work, data analysis, and conclusions are conducted and drawn by the human authors.

## B    APPENDIX

Table 6: Additional results on six benchmarks to demonstrate the effectiveness of VLSA. High-resolution benchmarks: MME-RealWorld Zhang et al. (2024c)(MME-Real), Q-bench-A1-single-dev Wu et al. (2023)(Q-bench); Hallucination evaluation benchmarks: Hallusion-Bench Guan et al. (2024)(Hall-Bench), AMBER-Discriminative Wang et al. (2023)(AMBER); Generalizability evaluation benchmarks: BLINK-Validation Fu et al. (2024)(BLINK), DEMON-Core Li et al. (2023f)(DEMON). "Avg-Acc" and "Acc" indicate "Average Accuracy" and "Accuracy", respectively.

| Method | High-resolution | | Hallucination | | Generalizability | |
|---|---|---|---|---|---|---|
| | MME-Real | Q-bench | Hall-Bench | AMBER | BLINK | DEMON |
| | (Avg-Acc) | (Avg-Acc) | (Avg-Acc) | (Acc) | (Avg-Acc) | (Avg-Acc) |
| LLaVA-1.5-HD | 26.1 | 58.6 | 44.4 | 75.3 | 36.9 | 24.6 |
| +VLSA | 30.3 | 59.1 | **49.2** | 82.5 | 38.7 | 26.6 |
| Δ | **+4.2** | **+0.5** | **+4.8** | **+7.2** | **+1.8** | **+2.0** |
| LLaVA-Next | 29.7 | 60.2 | 44.2 | 78.7 | 44.6 | 28.3 |
| +VLSA | **31.8** | **61.7** | 48.1 | **83.2** | **47.1** | **32.4** |
| Δ | **+2.1** | **+1.5** | **+3.9** | **+4.5** | **+2.5** | **+4.1** |

### B.1    IMPLEMENTATION DETAILS

As for the LLM backbone, we employ LLaMA3-8B-Instruct Dubey et al. (2024) for LLaVA-Next, Vicuna-7B-v1.5 Zheng et al. (2023) for LLaVA-1.5-HD. All of these LLMs feature transformer layers with a dimensionality of $d = 4096$. For the vision encoder, we utilize CLIP-ViT-L/14@336px Radford et al. (2021), which comprises 24 transformer layers, each with a feature dimension of $d = 1024$. The low-resolution snapshot $X_{\text{Lo}}$ referenced in Sec. 3.1 has dimensions of $\mathbb{R}^{3 \times 336 \times 336}$, aligning with the input image size required by the vision encoder. The length of the image embedding $V$, which is input to the LLM, consistently remains at 576, signifying a reduction of up to *five-fold* compared to the original LLaVA-Next and LLaVA-1.5-HD. The epigone described in Equation 4 is implemented as a low-rank MLP with two linear layers and the SiLU activation function. Its input and output dimensions are 4086, while the intermediate dimension is 256. Our development of VLSA builds upon the original LLaVA codebase with minor adjustments. We train all variants of VLSA using 8x NVIDIA A100 80GB GPUs with 500GB of system memory, deployed on the Volcengine cloud platform, completing each process within 19 hours. Table 7 summarizes the training data, trainable modules, and hyperparameters employed during each training stage. We implement greedy decoding with a temperature of 0 during inference to ensure reproducibility.

**VLSA is an efficient plug-in that seamlessly integrates into existing MLLMs:**

(1) During the training, our cognitive alignment functions as a data augmentation method conducted offline. The additional data labeling processes do not introduce any extra system overhead for training.

Table 7: Statistics of training data, trainable modules, and hyper-parameters.

| Stage | I | II |
|---|---|---|
| Training Data | LLaVA-558K-Pretrain | LLaVA-665K/790k-Finetune |
| Batch Size | 128 | 64 |
| Warm-up Ratio | 0.03 | 0.03 |
| Weight Decay | 0.00 | 0.00 |
| Trainable Module (learning rate) | perceiver (1e-3) Epigone (1e-3) Denoising Transformer (1e-3) | perceiver (1e-5) Epigone (1e-5) Denoising Transformer (1e-5) Vision Encoder (2e-6) Language Model (1e-5) |

Moreover, the SA-perceiver in perception alignment can easily replace the projector (such as MLP or Q-former) in other MLLMs, and the efficiency gained through compressive encoding offsets the extra overhead from reconstructive training (Please refer to Fig. 5).

(2) During the inference, all supplementary pretrained models (including VAE, Denoising Transformer, Epigone, and VQ-VAE) are deactivated. Consequently, VLSA does not increase the reasoning overhead.

**Benchmark selection:**

(1) To comprehensively compare our VLSA with LLaVA, we first follow LLaVA-1.5's dataset selection and evaluate competing methods on 8 academic-task-oriented datasets (including VQA$^{v2}$ Goyal et al. (2017), GQA Hudson and Manning (2019), VisWiz Gurari et al. (2018), ST-VQA Biten et al. (2019), ScienceQA-IMG Lu et al. (2022), TextVQA Singh et al. (2019), TextCaps Sidorov et al. (2020), COCO Chen et al. (2015)), and 7 instruction-following benchmarks that are tailor-made for MLLMs ( including POPE Li et al. (2023d), MME Fu et al. (2023), MMBench Liu et al. (2023c), MMBench-Chinese Liu et al. (2023c), SEED-Bench Li et al. (2023e), LLaVA-Bench (In-the-Wild) Liu et al. (2023a) and MM-Vet Yu et al. (2023)), in Tab. 1 and Tab. 2.

(2) Besides, we evaluate VLSA and LLaVA on 3 document understanding tasks (including AI2D Kembhavi et al. (2016), ChartQA Masry et al. (2022), and DocVQA Mathew et al. (2021) in Tab. 4) to assess their fine-grained understanding of rich visual information, in Tab. 4.

(3) Additionally, we further evaluate VLSA using benchmarks specifically designed for high-resolution images (MME-RealWorld Zhang et al. (2024c), Q-bench Wu et al. (2023)), benchmarks that assess hallucination (Hallusion-Bench Guan et al. (2024), AMBER Wang et al. (2023)), and benchmarks that test generalizability (BLINK Fu et al. (2024), DEMON Li et al. (2023f)) (Please refer to Appendix B.7).

## B.2    More Ablations

**Ablations on the structure of SA-Perceiver.** The SA-Perceiver is designed to integrate information from high-resolution images into the features of low-resolution images. Thus, we first employ a cross-attention layer to realize feature interaction. As high-resolution images are divided into multiple sub-images during preprocessing, we subsequently incorporate a self-attention layer after the cross-attention to enhance the modeling of the interrelationships among the aggregated sub-image information in low-resolution images' embeddings. To ensure parameter efficiency, we have omitted certain projection layers typically found in the standard attention mechanism. To demonstrate the effectiveness of our design choices, we conducted the following ablation experiments on the SA-Perceiver in Tab. 8: (ex1) removing the self-attention layer and only retaining the cross-attention layer. (ex2) retaining all linear projections in cross and self-attention, including those for generating keys, queries, values, and outputs.

Our rationale for introducing a learnable parameter $q$ as part of the input of the SA-Perceiver stems from the requirement of the text-to-image model, Stable Diffusion 3-medium, utilized in our reconstruction training, which necessitates a pooled embedding as input to encapsulate global semantic information. To better substantiate the efficacy of learnable parameter P, we have included

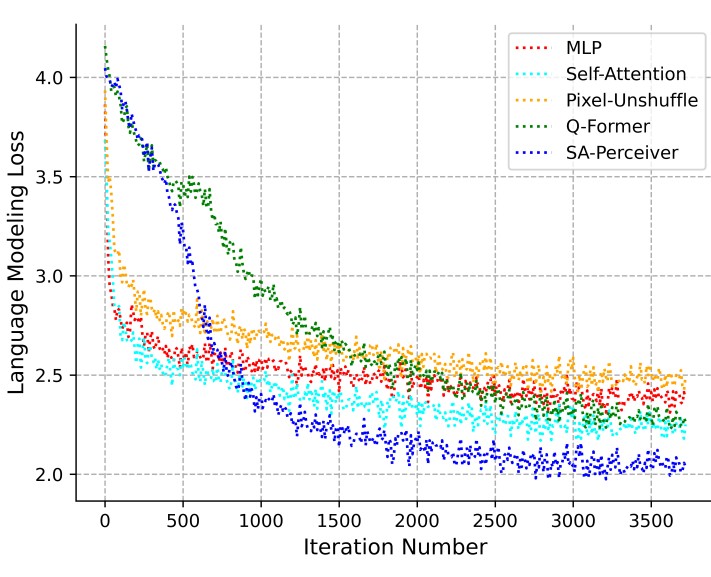

Figure 6: The influence of projectors on language modeling loss.

an ablation study (ex3) employing global pooling on the outputs of SA-Perceiver to generate the pooled embedding.

Table 8: Ablations on the structure of SA-Perceiver.

| Variant | GQA | SQA-I | DocVQA |
|---|---|---|---|
| (ex1) w/o. self-attn | 65.1 | 76.9 | 72.4 |
| (ex2) full projections | **65.5** | 77.3 | 75.0 |
| (ex3) global pooling | 64.9 | 76.2 | 73.3 |
| SA-Perceiver | 65.3 | **77.5** | **75.2** |

Table 9: Ablations on different visual projectors.

| Variant | GQA | SQA-I | DocVQA |
|---|---|---|---|
| (ex4) MLP | 62.1 | 70.1 | 73.9 |
| (ex5) Self-Attn | 62.3 | 69.6 | 74.1 |
| (ex6) Pixel-Unshuffle | 64.1 | 68.2 | 71.7 |
| (ex7) Q-former | 64.5 | 70.4 | 62.8 |
| SA-Perceiver | **65.3** | **77.5** | **75.2** |

To further illustrate the efficacy of our SA-Perceiver in handling high-resolution representations, we compare its performance against several visual projectors in Tab. 9. Specifically, we evaluate (ex4) an MLP or (ex5) a self-attention that processes high-resolution representations, (ex6) Pixel-Unshuffle Shi et al. (2016) combined with an MLP, where the pixel-unshuffle operation reduces the visual token count to one-quarter of its original value by resizing the tokens and concatenating them channel-wise, and (ex7) a Q-former that condenses high-resolution representations into a set of learnable queries with the same dimensionality as the low-resolution representations utilized in our SA-Perceiver. We also visualize their influence on language modeling loss during pretraining in Fig. 6.

**Ablations on the LDM.** To further substantiate the necessity of using text-to-image LDMs, we have conducted new ablations where we implement reconstruction training by replacing the SD with pretrained vision-only models (ex8) VAE decoder Kingma (2013), (ex9) VQ-VAE, and (ex10) MAE He et al. (2022) decodervan den Oord et al. (2018). The results are reported in Tab. 10. Besides, we validate the performance change when substituting the current LDM with (ex11) stable-diffusion-2-small Rombach et al. (2022), a version that has fewer parameters, and (ex12) randomly initialized stable-diffusion3-medium trained from scratch. We find that VLSA shows low sensitivity to the scale of LDM, which is important for further adapting VLSA to resource-limited environments.

Table 10: Ablations on the image generation module.

| Variant | GQA | SQA-I | DocVQA |
|---|---|---|---|
| (ex8) VAE | 64.7 | 75.0 | 74.4 |
| (ex9) VQ-VAE | 65.2 | 74.2 | 69.3 |
| (ex10) MAE | 64.5 | 74.3 | **75.4** |
| (ex11) stable-diffusion2-base | 64.9 | 77.2 | 75.1 |
| (ex12) stable-diffusion3-medium (from scratch) | 65.0 | 76.3 | 74.7 |
| stable-diffusion3-medium | **65.3** | **77.5** | 75.2 |

Table 11: Ablations on the image encoder.

| Variant | Vision Encoder | Batchsize | AI2D | ChartQA | DocVQA |
|---|---|---|---|---|---|
| LLaVA-Next (Reported) | SigLIP | 512 | 71.6 | 69.5 | 78.2 |
| (ex13) LLaVA-Next (Reproduced) | SigLIP | 128 | 71.4 | 69.2 | 78.3 |
| (ex14) LLaVA-Next + VLSA | SigLIP | 128 | **73.7 (+2.3)** | **71.2 (+2.0)** | **81.1 (+2.9)** |
| LLaVA-Next | CLIP | 128 | 69.5 | 67.1 | 75.2 |
| LLaVA-Next + VLSA | CLIP | 128 | 71.4 (+1.9) | 67.9 (+0.8) | 75.2 (+1.5) |

**Ablations on the image encoder.** We evaluate the influence of (ex13) employing the stronger vision encoder SigLIP Zhai et al. (2023) in Tab. 11. Note that there are discrepancies between the results of (ex14) our reproduced LLaVA-Next(SigLIP) and those reported officially. Due to limitations in computational resources, we have to use a smaller equivalent batch size, which leads to a slight shift in performance.

**Ablations on the language model.** We report the performance of VLSA with the replacement of the language backbone to (ex15) Vicuna1.5-7B, (ex16) Vicuna1.5-13B, and (ex17) Qwen1.5-72B. Additionally, we report the performance of LLaVA-Next with these backbones as references.

Table 12: Ablations on the language model

| Variant | LLM | AI2D | ChartQA | DocVQA |
|---|---|---|---|---|
| LLaVA-Next | Vicuna1.5-7B | 66.4 | 54.7 | 72.5 |
| (ex15) LLaVA-Next + VLSA | Vicuna1.5-7B | 67.5 (+0.9) | 55.3 (+0.6) | 74.6 (+2.1) |
| LLaVA-Next | Vicuna1.5-13B | 67.0 | 60.5 | 75.7 |
| (ex16) LLaVA-Next + VLSA | Vicuna1.5-13B | 69.2 (+2.2) | 61.9 (+1.4) | 79.8 (+4.1) |
| LLaVA-Next | Qwen1.5-72B | 73.4 | 72.7 | 79.9 |
| (ex17) LLaVA-Next + VLSA | Qwen1.5-72B | 77.1 (+3.7) | 77.3 (+4.6) | 82.5 (+2.6) |

**Ablations on the loss ratio.** We evaluate the influence of the different ratios between the reconstruction loss $\mathcal{L}_{rec}$ and the language modeling loss $\mathcal{L}$ in Tab. 13.

Table 13: Ablations on the loss ratio.

| Variant | AI2D | ChartQA | DocVQA |
|---|---|---|---|
| (ex18) 0.25: 0.75 | 71.0 | 67.2 | 74.4 |
| (ex19) 0.5: 0.5 | 70.1 | 67.5 | 73.9 |
| (ex20) 0.75: 0.25 | 68.2 | 67.6 | 75.1 |
| 1: 1 | **71.4** | **67.9** | **75.2** |

**Ablations on the cognition alignment.** To assess the effectiveness and universality of our SSFT objectives in cognitive alignment, we compare them with previous techniques that enhance the LLM's understanding of visual semantics. In particular, we examine (ex21) the additional visual grounding objectives by incorporating the grounded visual chat (GVC) dataset from LLaVA-Grounding Zhang et al. (2023b) into LLaVA-Next's 790K fine-tuning instances, and (ex22) the additional segmentation objectives achieved by integrating LISA Lai et al. (2024). All experiments presented in Tab. 14 utilize the same CLIP encoder as employed in our other experiments. Additionally, we exclude perception alignment (which includes compressive encoding and reconstructive training) for VLSA to ensure fair comparisons.

Table 14: Ablations on the effectiveness of Cognition Alignment.

| Variant | AI2D | ChartQA | DocVQA |
|---|---|---|---|
| (ex21) visual grounding | 68.7 | 67.1 | 73.0 |
| (ex22) segmentation | 68.5 | 66.8 | **74.4** |
| pixel value+codebook indices | **72.5** | 67.0 | 73.9 |

### B.3 ANALYSIS OF EFFICIENCY EVALUATION

In Fig. 5, we compare the latency and Flops of the baseline model and VLSA during the feedforward process of processing 1,000 images across various input resolutions. Despite adding multiscale visual feature interactions in SA-Perceiver and integrating reconstructive training, our VLSA still showcases impressive efficiency. Below, we outline two potential reasons that may contribute to this phenomenon.

(1) As for SA-Perceiver, which comprising four $\mathbb{R}^{1024 \times 1024}$ linear layers and one $\mathbb{R}^{1024 \times 4096}$ linear layer (save 60% parameters compared with the projection module in LLaVA-Next, which consists of a $\mathbb{R}^{1024 \times 4096}$ and a $\mathbb{R}^{4096 \times 4096}$ linear layer), to integrate high-resolution image information into low-resolution image features at a lower cost. Only the low-resolution features are then utilized as input to the LLM. Given that the projection module (including MLP, Q-former, and our SA-Perceiver) has a significantly lower parameter count and computational complexity than the LLM, the overall system latency is predominantly dictated by the computation delay of the LLM. As SA-Perceiver enables a reduction of visual sequence length up to five-fold, and the time complexity of LLM is $O\left(n^2\right)$, our method can theoretically achieve a maximum reduction in latency by a factor of 25. (However, system latency is also affected by factors such as the number of input text tokens, the length of the generated sentences, and other intricate system dynamics.)

(2) During our reconstructive training, the LDM is required to perform only a single denoising step per iteration, in contrast to the multi-step denoising process used for image generation. Consequently, its computational overhead is much lower than the feedforward process of the LLM, and the additional costs brought by LDM are substantially outweighed by the efficiencies gained through our compressive encoding.

### B.4 MORE EVALUATIONS ON MODALITY ALIGNMENT

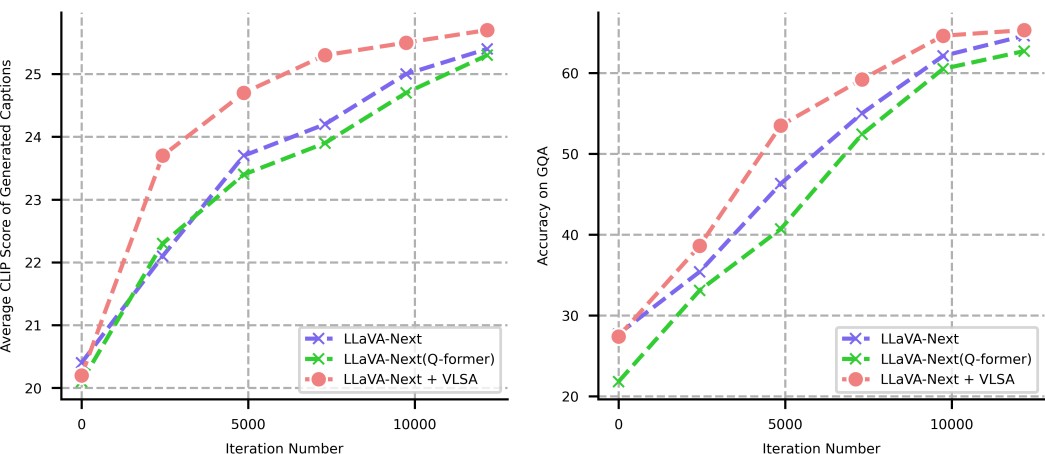

Figure 7: **(Left)** Changes in average CLIP Score during training. **(Right)** Changes in accuracy on GQA benchmark during training.

We quantitatively evaluate the efficiency of VLSA in facilitating modality alignment. Specifically, we trained three variants (LLaVA-Next, LLaVA-Next with Q-former as the projector, and LLaVA-Next integrated with our VLSA) using 790k visual instruction tuning data, saving a checkpoint every 2435 iterations (one epoch consisting of 12,176 iterations). We then assessed the quality of modality alignment using the following two metrics, with higher values indicating better alignment:

**(1) CLIP Score:** We randomly selected 1,000 images for each checkpoint and prompted the model to generate detailed captions, then employed the CLIP model to evaluate the correlation logits between the input images and their corresponding generated captions. The final score was determined by calculating the average of these logits. Results are shown in Fig. 7 (Left).

**(2) Benchmark Performance:** For each checkpoint, we test their performance on the GQA benchmark. Results are shown in Fig. 7 (Right).

Table 15: Language modeling losses on the image captioning task based on codebook indices.

| Epoch | 0.2 | 0.4 | 0.6 | 0.8 | 1.0 |
|---|---|---|---|---|---|
| (ex23) LLaVA-Next + VLSA (w/o SSFT) | 1.94 | 1.29 | 1.12 | 0.93 | 0.88 |
| LLaVA-Next + VLSA | 0.73 | 0.62 | 0.57 | 0.52 | 0.54 |

These evaluations enable us to validate the impact of our proposed method on modality alignment by analyzing the variations in model performance throughout the training process. The results demonstrate that VLSA achieves modality alignment more quickly and effectively.

### B.5 THE EFFECTIVENESS OF PREDICTING CODEBOOK INDICES

We explore the rationale underpinning the effectiveness of predicting codebook indices. Pertaining to the assertion articulated in Sec 3.2, the mapping between natural language and codebook indices may originate from the multi-task learning and transfer learning capabilities inherent in LLMs. This suggests that LLMs can utilize visual semantics as intermediate representations, thereby facilitating the connection between natural language and codebook indices. To substantiate this claim, we propose additional experiments: specifically, we fine-tune the well-trained VLSA (integrated into LLaVA-Next), alongside its variant that omits cognition alignment, on a task that necessitates generating captions based on codebook indices rather than direct image inputs. As shown in Tab. 15, the comparison reveals that VLSA demonstrates a significantly lower loss relative to its variant, suggesting that the prediction of codebook indices indeed enhances the mapping between natural language and Codebook indices. Meanwhile, codebook indices prediction aids LLMs in comprehending the relationship between visual semantics and codebook indices, ultimately bolstering LLMs' capacity to correlate visual semantics with natural language.

### B.6 LANGUAGE MODELING LOSS

For each instruction in the augmented dataset, we encode it into word token embeddings $I_{emb}$, which are then concatenated with image embeddings $V$, serving as inputs for the LLM to generate a response $R$, which is formulated as:

$$R = g_{LLM} \left( [V,\ I_{emb}] \right), \tag{6}$$

where $g_{LLM}$ symbolizes the LLM's reasoning process. The optimization of language modeling can be formulated as:

$$\arg\min \mathcal{L} \left( R, Target;\ \theta_{LLM} \right). \tag{7}$$

Here, $Target$ refers to the augmented ground-truth answer, $\mathcal{L}(\cdot)$ denotes the language modeling loss, and $\theta_{LLM}$ represents the LLM's parameters. $\mathcal{L}(\cdot)$ is defined as:

$$\mathcal{L} = \sum_{i=1}^{B} \sum_{j=1}^{K+1} \log p \left( y_j^i | V^i,\ I_{emb}^i,\ Target_{0:j-1}^i;\ \theta_{LLM} \right), \tag{8}$$

where $B$ denotes the batch size, and $K$ is the length of the response $R$. The final loss for the entire system is determined through the summation of $\mathcal{L}$ and $\mathcal{L}_{rec}$.

### B.7 ADDITIONAL QUANTITATIVE RESULTS

We further assess the effectiveness of integrating VLSA into existing MLLMs across six carefully curated datasets, in addition to those discussed in Sec. 4. As illustrated in Table 6, we initially focus on benchmarks tailored for high-resolution images, including MME-RealWorld Zhang et al. (2024c) and Q-bench-A1-single-dev Wu et al. (2023). The enhancements brought about by VLSA suggest that our method can more effectively leverage the rich visual information available in high-resolution inputs. Additionally, we have selected two benchmarks specifically designed to evaluate hallucinations in MLLMs: Hallusion-Bench Guan et al. (2024) and AMBER-Discriminative Wang et al. (2023). The substantial performance improvements achieved with VLSA highlight its capability to efficiently minimize information loss in visual inference and alleviate hallucinations. Furthermore, beyond conventional benchmarks (those involving single-image inputs with textual outputs), we validate the versatility of VLSA by testing it on BLINK-Validation Fu et al. (2024) and DEMON-Core Li et al. (2023f). These datasets involve multiple, interleaved, and multimodal instructions, showcasing the

required context to complete various tasks. The consistent performance gains confirm that VLSA serves as a general-purpose solution for enhancing visual-semantic understanding.

### B.8 COMPARISON WITH NATIVE MULTIMODAL FRAMEWORKS

To gain a deeper understanding of VLSA, we can compare it with native multimodal models such as Chameleon Team (2024), Show-o Xie et al. (2024b), and Janus Chen et al. (2025b):

VLSA operates as a multimodal model centered around the LLM, enabling the LLM to comprehend visual inputs through a limited set of multimodal data. The core of VLSA lies in its efficient pursuit of modality alignment between vision and language. In contrast, native multimodal models like Chameleon necessitate a fundamental update to their LLM-parameter-initialized multimodal decoders using extensive multimodal datasets, although they possess the capability to perform multimodal understanding and generation tasks concurrently.

From an architectural perspective, the LLM decoder in VLSA processes both textual and visual features that have been refined through projectors, including MLP, Q-Former, and our SA-Perceiver. In contrast, multimodal decoders in models such as Chameleon employ unaligned visual and textual features. This difference likely accounts for the higher data demands of the latter, as they must adapt their decoders to a shifted distribution, despite generally having greater potential.

Regarding training objectives, VLSA aims to 1) minimize information loss incurred by the visual encoder and projectors through reconstructive training and 2) enhance the LLM's understanding of visual semantics via an unimodal autoregressive task, which predicts the textualized discrete visual semantic labels (codebook indices) of images. Collectively, these objectives optimize modality alignment. On the other hand, models like Chameleon enable decoders to directly process visual features through tasks involving image reconstruction (generation) or multimodal autoregression.

### B.9 COMPARISON WITH EARLY FUSION METHODS

Regarding QA-ViT Ganz et al. (2024), EMMA Ghazanfari et al. (2025), BRAVE Kar et al. (2024), and mPLUG-Owl2 Ye et al. (2023) mentioned in Sec 2: These works adopt a similar approach to enhancing modality alignment by modifying the location of Visual-Language (VL) feature fusion. Conceptually, methods like LLaVA, Qwen-VL, and InternVL series, which fuse VL features only within the LLM, are late-fusion. The four mentioned works allow preliminary VL feature interaction before input to the LLM, termed early fusion:

(1) QA-ViT: Implements VL early fusion by adding cross-modal attention layers within the visual encoder.

(2) EMMA: Uses the "modality adaptation" module based on linear projections for VL early fusion.

(3) BRAVE: Fuses features from multiple visual encoders with text features in an improved Q-Former (MEQ-Former) for VL early fusion.

(4) mPLUG-Owl2: Employs a similar idea to EMMA, but first uses a compression module to reduce visual sequence length and employs a more complex cross-attention-based modality adaptation module.

Compared to late fusion, early fusion better filters visual information to focus on user requests, boosting performance on standard VQA tasks. However, this approach is a double-edged sword: performance degrades when user queries are ambiguous/overly verbose (e.g., SQA) or during fine-grained perception tasks (e.g., MME, ChartQA, InfoVQA). This likely explains why top-performing MLLMs like Qwen2.5-VL Bai et al. (2025) and InternVL-2.5 Chen et al. (2025a) avoid early fusion. The table below reports replication results for QA-ViT, showing significant performance drops versus baseline on SQA-I (containing verbose lectures hinting at solutions) and MME & InfoVQA (requiring fine-grained perception).

In contrast, our VLSA improves modality alignment from a different perspective: explicitly constraining information loss during the reasoning. (Early fusion methods can be seen as reducing the harm of lost information based on text-visual relevance, but cannot directly minimize the information loss.) As demonstrated in the paper, VLSA delivers consistent performance gains across diverse downstream tasks without harming specific capabilities, making it more generalizable. Crucially,

Table 16: Comparison of the early fusion method with the baseline.

| Method | VQA$^{v2}$ | GQA | MM-VeT | InfoVQA | COCO-cap | SQA-I | MME |
|---|---|---|---|---|---|---|---|
| (ex24) LLaVA-1.5+QA-ViT | **77.8** | **62.9** | **31.6** | 25.5 | 109.3 | 66.6 | 1394 |
| LLaVA-1.5 | 76.6 | 62.0 | 30.6 | **26.1** | **109.4** | **69.8** | **1463** |

VLSA seamlessly integrates with existing MLLM architectures. (We have validated VLSA in LLaVA-Next, MiniGemini, SliME, Qwen2.5-VL, and InternVL2.5.) Particularly for dynamic high-resolution methods that already employ visual feature compression (e.g., Qwen-VL, InternVL2.5), VLSA can omit SA-Perceiver, integrate Reconstructive Training and Cognition Alignment into their training phase to enhance performance further. Critically, during deployment, this integration introduces no structural changes and zero computational overhead to the base model.

### B.10 COMPARISON WITH EXISTING RECONSTRUCTIVE TRAINING OBJECTIVES.

The key distinction between concurrent work Ross Wang et al. (2024b) and VLSA is the source of reconstruction cues: Ross uses image tokens processed by the LLM, while VLSA uses the output of the alignment projector. (We also initialize the denoising module with a pre-trained text-to-image LDM vs. Ross's random initialization.) Indeed, the Ross-style structure was considered as one of the alternatives during the development of VLSA. Although using the LLM's output as the reconstruction cue can also help mitigate information loss, our experiments demonstrated that its performance was inferior to the solution we adopted:

Table 17: Comparison of different clues for reconstruction.

| Variant | GQA | SQA-I | DocVQA |
|---|---|---|---|
| (ex25) LLM's outputs as clues (Ross-style) | 62.9 | 75.1 | 75.0 |
| projector's outputs as clues (ours) | 65.3 | 77.5 | 75.2 |

Two plausible reasons can be identified for this performance discrepancy. First, as discussed in Sec 3.1, using the projector's outputs as reconstruction cues leverages the pre-trained knowledge of the text-to-image LDM, enhancing the semantic alignment between visual and textual tokens before input into the LLM. The stronger this alignment, the more effectively the LLM can process the visual information that visual tokens contain. Second, the Ross-style structure necessitates the LLM learning to convey visual information. When optimized alongside our cognition alignment objectives, this learning objective may encounter challenges related to multi-objective optimization. Additionally, our further research on VLSA reveals that the Ross structure causes noticeably more severe forgetting on text-only tasks after visual instruction tuning (since this issue lies outside the scope of this paper, we will not discuss it in detail here).

### B.11 COGNITION ALIGNMENT TEMPLATES

We report lists of instructions used to predict high-level and low-level visual semantics in Tab. 19 and Tab. 18. They present the same meaning with natural language variance. For each input image-conversations pair in the 790K visual instruction tuning dataset from LLaVA-Next Liu et al. (2024b), we randomly sample a template from each list and expand it with information on the current image's dimensions and the region we care about. For templates predicting high-level visual semantics, we also provide a hint regarding the length of the targeted VQ indices based on the specific VQ-VAE used. The expanded templates are then combined with $Target_{\text{VQ}}$ and $Target_{\text{PX}}$ in Sec. 3.2 to create additional conversations, which are inserted at random positions within the original conversations. We report one example used in visual instruction tuning with cognition alignment in Tab. 20.

### B.12 MORE CASE STUDIES

We provide more comparisons between LLaVA-Next before and after integrating our VLSA on visual writing tasks in Fig. 9 and Fig. 10, and on visual question answering tasks in Fig. 8. These qualitative results substantiate the effectiveness of VLSA in reducing information loss.

Table 18: The list of instructions for predicting pixel RGB values of images.

- For each pixel in this image, please provide the corresponding RGB value.
- I would like to know the exact RGB value of every pixel depicted in this image.
- Could you please list the RGB values for all pixels in this image?
- Please give me a detailed breakdown of the RGB values for each pixel in this image.
- I need the RGB value of each and every pixel in this image, can you help?
- Would it be possible to obtain the RGB values for all the pixels in this image?
- Please provide a comprehensive report on the RGB values of each pixel in this image.
- I'm looking for the RGB value of each pixel in this image, could you assist?
- Can you please specify the RGB value for every single pixel in this image?
- I require the RGB values of all pixels in this image, please provide them.
- Please tell me the RGB value of each pixel in this image, from top left to bottom right.
- I'm interested in the RGB values of each pixel, can you give me that information for this image?
- Could you please share the RGB values for each pixel in this image with me?
- I would appreciate it if you could tell me the RGB value of each pixel in this image.
- Please provide the RGB values for all pixels in this image, in a readable format.
- I'm seeking the RGB value of each pixel in this image, can you help me out?
- Would you be able to give me the RGB values for each pixel in this image?
- I need to know the RGB value of every pixel in this image, please provide the details.
- Please tell me the RGB value of each and every pixel that makes up this image.
- I'm looking to obtain the RGB values for all pixels in this image, can you assist me?

Table 19: The list of instructions for generating VQ indices of images.

- Provide the VQ indices that would represent the essential features of this image, capturing its key visual elements.

- Derive the VQ indices that capture the diversity and complexity of this image's visual information.

- Describe the VQ indices that effectively represent the visual essence of this image.

- Assuming a VQ-based representation, what are the indices that would summarize the key aspects of this image?

- What are the VQ indices that would allow for the reconstruction or recognition of this image, based on its most salient features?

- Imagine the VQ indices as a codebook of visual elements, and devise the sequence that best represents the narrative or story within this image.

- What are the VQ indices that summarize the most prominent shapes, lines, and forms found within this image, forming a visual abstract?

- Formulate the VQ indices as a set of vectors that represent the hierarchical organization of visual information in this image, from general to specific.

- Assuming a VQ-VAE model, what would the latent space representations (VQ indices) of this image look like, encoding its generative factors?

- Describe the VQ indices that would enable a machine learning model to recognize or classify this image based on its visual characteristics and patterns.

Table 20: One example used in visual instruction tuning with cognition alignment.

**Conversations:**

● from: Human

<image> What are the colors of the bus? Answer the question with gpt-T-COCO format.

● from: GPT

The bus in the image is white and red.

============================================================

● from: Human

What feature can be seen on the back of the bus?

● from: GPT

The back of the bus features an advertisement.

============================================================

● from: Human

Imagine a hypothetical encoding process for the upper left quarter of this image. What would the resulting VQ indices look like, summarizing its content? Note that the image resolution is [3, 672, 672], so you should generate 1764 indices

● from: GPT

<vq_start>3492, 3492, 532, 532, 532, 532, 532, 532, 532, 532, 532, 532, 532, 532, 532, 532, 532, 532, 532, 532, 532, 532, 532, ......, 2054, 3764, 4015, 570, 2675, 344, 3369, 580, 2413, 282, 1653, 2826, 1593, 2253, 1988, 3105, 3488, 1455, 1945, 3069, 1118, 1130, 1130, 2571, 2883, 3160, 1433, 3428, 815, 901]<vq_end>

============================================================

● from: Human

Is the bus driving down the street or pulled off to the side?

● from: GPT

The bus is driving down the street, which is crowded with people and other vehicles.

============================================================

● from: Human

Please tell me the RGB value of each and every pixel that makes up the upper left quarter of this image. Note that the image resolution is [3, 672, 672].

● from: GPT

<rgb_start>[[205, 203, 191], [203, 201, 189], [202, 200, 188], [183, 183, 171], [208, 209, 195], [186, 183, 166], ..., [205, 203, 191], [204, 202, 190], [203, 201, 189]]<rgb_end>

Figure 8: VLSA reduces the loss of visual information and mitigates hallucinations in VQA tasks.

**User:**

Examine the image closely and identify as many details as you can. Pay attention to landmarks, architectural features, natural elements, signs, and any other clues that might help determine the location. Based on these details, tell me where this place is.

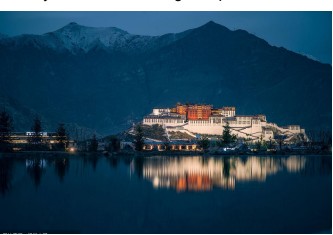

**LLaVA-Next**

The image shows a large, multi-story building with a traditional architectural style, featuring multiple levels and a flat roof. The building is illuminated, suggesting it is a significant structure, possibly a landmark or a place of interest. It is situated on the edge of a body of water, which reflects the building's lights and the surrounding landscape. The reflection is visible on the water's surface, indicating that the water is calm. In the background, there are snow-capped mountains, suggesting a mountainous region. The sky is dark, which could indicate either dusk or dawn, or it could be a cloudy day. There are no visible texts or inscriptions in the image. The style of the image is a photograph with a focus on the building and its surroundings."

**LLaVA-Next + VLSA**

The image depicts the **Potala Palace**, a renowned landmark located in Lhasa, Tibet. The palace is a large, multi-tiered structure with a distinctive red and white color scheme. The building has a series of rectangular sections with flat roofs, and the facade appears to be made of **stone or concrete**. The structure is situated on a hillside, and there are **trees in the foreground**, reflecting on the water below. It is surrounded by a serene lake and snow-capped mountains in the background. The palace is illuminated at night, casting a **warm glow** that contrasts with the cool blue of the sky. The image is a photograph, and there are no visible texts or distinctive brands that provide additional information about the location. The Potala Palace is a significant cultural and historical site, known for its grandeur and architectural beauty.

Figure 9: VLSA provides more accurate object recognition and detailed descriptions.

**User:**

Compose an engaging story about this picture, explaining where this place is and highlighting the animals and their activity.

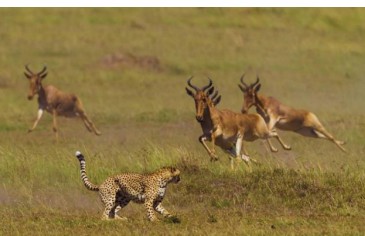

**LLaVA-Next**

In the vast expanse of the African savanna, a young **leopard** is on the hunt. The leopard, with its agile body and sharp senses, is crouched low to the ground, stalking its prey. It is in pursuit of a group of three antelopes.

The presence of the leopard indicates that this area is rich in prey, making it a suitable habitat for predators like the leopard. The animals' activity of running and chasing each other showcases the natural behavior and interactions between these species in their shared environment.

**LLaVA-Next + VLSA**

In the vast expanse of the African savanna, a group of three antelopes are captured in a moment of pure exhilaration. They are running across **a grassy field**, their **hooves pounding the earth** as they are chased by a **cheetah**. The cheetah, with its distinctive **black and yellow spots**, is in the foreground of the image, its body low to the ground as it **dashes towards the right side** of the frame. The antelopes, with their long, **curved horns**, are in hot pursuit. They are spread out across the field, their bodies leaning forward in an attempt to escape from the cheetah. The savanna stretches out behind them, a sea of green grass. The scene is a snapshot of life in the wild, a testament to the beauty and power of nature.

Figure 10: VLSA provides more detailed, visually-grounded responses than LLaVA-Next.