# OpenReview forum: "VLSA: Enhancing Vision-Language Understanding via Perception and Cognition Alignment"
_ICLR.cc/2026/Conference — Submitted to ICLR 2026_

### Official Review · Reviewer_sE3s · 2025-10-27

**Soundness:** 3
**Presentation:** 3
**Contribution:** 3
**Rating:** 4
**Confidence:** 4

**Summary:**

This paper presents VLSA, which reframes multimodal alignment as reducing information loss along the vision→language pipeline. The method has two stages: Perception Alignment that injects high-res patch details into a low-res structural stream, paired with reconstruction training (LDM-based) to mitigate encoding loss; and Cognition Alignment, where the LLM predicts VQ-VAE code indices and sampled pixels to lessen decoding loss. In inference, auxiliary modules are disabled; the visual token length is fixed, yielding both accuracy and efficiency gains across many benchmarks and several MLLM backbones.

**Strengths:**

1） Clear two-stage framework with practical appeal. The design explicitly targets perception-side and cognition-side loss, is plug-and-play, and keeps inference lightweight.

2）Good empirical coverage and portability. The approach delivers consistent gains across diverse tasks and multiple MLLMs, suggesting robustness of the alignment recipe.

**Weaknesses:**

1）Motivation–method disconnect. Although framed under “information-loss reduction,” the three components, compressed high-res encoding, LDM-based reconstruction, and SSFT, are presented more as parallel tricks than a tightly coupled mechanism. The paper does not convincingly argue why all three are necessary together, how their gradients/targets complement rather than compete, or how each specifically addresses the stated losses beyond high-level intuition. This reads as a stitched trio rather than a single, strongly motivated design.

2）Compressed high-res input is not novel; conflict-avoidance remains unclear. Multi-resolution schemes that mix high-res patches with a low-res global stream have prior art. The paper should clearly articulate how SA-Perceiver prevents conflicts between high-res local details and low-res global semantics during injection. What controls (gating, residual scaling, normalization, attention head specialization) ensure injected details don’t overwhelm global semantics? Any controlled comparisons that show an optimal regime and failure modes?

3）Key hyperparameters lack guidance. The paper sets defaults but does not provide sensitivity/selection rationale for crucial knobs (k, τ, λ₁, λ₂, δ). Even a brief stability range or small sensitivity curves/tables would improve clarity for practitioners.

4）Ablation granularity on cross-component synergy is limited. Existing ablations mostly compare within components. To support the “triad” motivation, the paper should include cross-component causal ablations under a fixed budget: only compressed encoder, +reconstruction, +SSFT, full, with convergence curves and analysis of additive vs. synergistic gains.

**Questions:**

Please refer to Weakness

---

> ### Author Response · Authors · 2025-11-21
>
> We sincerely appreciate the time and effort you have dedicated to reviewing our manuscript. In the following sections, we have made a concerted effort to address your concerns in detail. We hope our responses will elucidate the contributions and significance of our research.
> ## w1:
> Our motivation is to enhance model performance by minimizing information loss during the forward process of MLLMs. As outlined in Lines 46-53, we categorize the total information loss into three distinct stages: Original Information Loss (O-IL), Encoding Information Loss (E-IL), and Decoding Information Loss (D-IL). In response, we propose: (a) compressive high-resolution encoding to address O-IL without incurring additional computational costs, (b) LDM-based reconstruction to alleviate E-IL while fostering VL alignment, and (c) SSFT objectives to reduce D-IL by strengthening MLLMs' overall comprehension of visual features. **Each of these three components targets a specific type of information loss and is designed to be complementary**—only through the combined application of all three can we comprehensively alleviate information loss in MLLMs' forward process.
>
> The paper elaborates on the design motivation, intuition, and specific methodology for each component that addresses the corresponding information loss (please refer to Lines 100-117 and Section 3), and presents **comprehensive ablations in the experimental section, illustrating the individual effectiveness, limitations, and the importance of their integration**. For a thorough analysis, please refer to Table 4 and Section 4.2. To save the reviewer's time, we have summarized some key arguments and conclusions below:
>
> 1. Compared to computationally expensive high-resolution methods, naively downsampling input images to low resolution degrades performance on perception-oriented benchmarks like OCR, but significantly reduces computational cost and improves performance on tasks requiring global understanding, such as high-level VQA.
> 2. Replacing naive downsampling with (a) compressive high-res encoding preserves the global understanding capability of low-resolution methods while substantially narrowing the gap with high-resolution methods in detail perception, demonstrating its effectiveness in mitigating O-IL while maintaining computational efficiency.
> 3. The combination of (a) and (b) through LDM-based reconstruction further enhances detail perception, even surpassing high-resolution methods. This underscores the effectiveness of (b) in mitigating E-IL (as it impacts only the visual encoder without altering the input image size or the LLM). However, this enhancement in local detail within the visual features fed into the LLM may complicate comprehension, potentially leading to a decline in global understanding (high-level VQA tasks) when compared to using (a) alone.
> 4. Combining (a), (b), and (c) SSFT objectives yields stable improvements in both detail perception and global understanding compared to high-resolution methods, demonstrating the effectiveness of (c) in mitigating D-IL and the necessity of combining all three components.
>
> **Note:** Enabling (a) compressive high-res encoding is a prerequisite for applying (b) LDM-based reconstruction and (c) SSFT objectives. When the input image size is too large, applying (b) or (c) alone would incur computational costs exceeding naive high-resolution methods (unacceptable under resource constraints), violating VLSA's design principle of computational efficiency.
>
> ## w2:
> (1) Regarding the novelty of compressive high-resolution encoding, although this is an active area of research, we discuss pertinent multi-resolution scheme works in the Related Works section (Lines 129-139) and highlight their limitations. This paper introduces SA-Perceiver, an efficient multi-resolution feature aggregation module within our Perception Alignment pipeline, specifically designed for models that lack compressive encoding capabilities (such as the LLaVA Series). For models that already possess similar functionalities (such as Qwen2.5-VL in our experiments), we can omit SA-Perceiver and preserve their original designs. Furthermore, we compare SA-Perceiver with commonly used multi-resolution schemes (Q-former, Pixel-Unshuffle, self-attention, MLP) in Appendix Figure 6 and Table 9, demonstrating its effectiveness and significance.

---

> ### Author Response · Authors · 2025-11-21
>
> (2) Conflicts between local details and global semantics: The reviewer raises a profound and thought-provoking question. In our understanding, local details and global semantics are not inherently at odds; their apparent "contradiction" likely arises from the LLM's difficulty in effectively comprehending and utilizing these features, particularly when faced with abundant local details. Our VLSA incorporates two key designs to address this challenge:
>
> 1. As illustrated in Figure 3, SA-Perceiver utilizes Low-Resolution Embeddings, which exhibit more "global" characteristics, as Queries. It aggregates information from High-Resolution Embeddings through cross-attention, effectively organizing local details within the structure of global semantics. **This method of information extraction can be regarded as an inductive bias of SA-Perceiver that enhances its ability to handle local details.** As noted in Lines 216-217, compared to prior approaches, this design alleviates the challenge of modeling visual content—especially content rich in details—by avoiding relying solely on causal attention within LLMs. Our ablations (Figure 6 and Table 9) demonstrate that this approach yields better convergence and improves performance on downstream tasks when compared to methods that simply concatenate multi-resolution embeddings before processing (Self-Attention, MLP, and Q-former). These evidences show that SA-Perceiver allows the model to effectively leverage both local details and global semantics simultaneously.
> 2. The incorporation of SSFT objectives enhances the LLM's ability to comprehend both local details and global semantics. Along with Reconstructive Training, **these objectives enable SA-Perceiver to automatically "balance" the significance of these two types of information during the end-to-end training process.**
>
> We thank the reviewer for this insightful question and commit to presenting these discussions in the revised manuscript.
>
> ## w3:
> Since this paper focuses more on validating VLSA as an efficient performance-enhancing plugin for various baseline models rather than providing a highly optimized MLLM, **we ensure all hyperparameters used in experiments are consistent with baselines to guarantee fairness**, without conducting further ablations on related hyperparameters (additionally, some hyperparameters the reviewer mentioned are not involved in the paper). However, we acknowledge that current settings may not be optimal, and more reasonable hyperparameter selection could further improve performance.
>
> ## w4:
> We primarily carry out cross-component ablation studies in the main text, and ablations within individual components are discussed in the Appendix. For details, please refer to Table 4 and Lines 445-468, where **we illustrate the effectiveness and limitations of each component when utilized independently, as well as the necessity and synergistic benefits of their combination.** To facilitate understanding, we also summarize key analyses and conclusions in **response w1**. Furthermore, Figures 6 and 7 present convergence curves for VLSA and the baseline models, showcasing language modeling loss and performance metrics for downstream tasks, respectively.

---

### Official Review · Reviewer_1pvz · 2025-10-31

**Soundness:** 3
**Presentation:** 3
**Contribution:** 3
**Rating:** 6
**Confidence:** 4

**Summary:**

The paper reframes VL alignment as reducing information loss along the vision→language path and proposes a plug-in, two-stage method: Perception Alignment (compressive SA-Perceiver plus reconstructive training with an LDM so that embeddings (V,P) can recover the input) and Cognition Alignment (SSFT that has the LLM predict VQ-VAE codebook indices and pixels as auxiliary targets). The approach is trained in two stages and evaluated across 7 architectures and 25+ benchmarks; at inference the visual sequence is fixed at 576 tokens and all auxiliary models are disabled.

**Strengths:**

1. Clear, principled decomposition. Casting hallucination as information-loss (encoding/decoding) motivates the dual alignment design with concrete mechanisms and objectives.

2. Reconstruction-driven alignment. Using an LDM to enforce recoverability from (V,P) is well justified and supported by ablations on alternative decoders.

3. Practical efficiency. A constant 576-token visual sequence and deactivated auxiliaries at test time.

4. Broad applicability with measurable gains. Improvements shown on LLaVA variants and other MLLMs.

**Weaknesses:**

Limited diagnosis of why and when VLSA helps. Although ablations are included (SA-Perceiver design, LDM variant, encoders/backbones, loss ratio), the paper lacks error taxonomies explaining failure modes.

**Questions:**

Please refer to weaknesses.

---

> ### Author Response · Authors · 2025-11-21
>
> We would like to thank the reviewers for the recognition of our work.
>
> Below, we will elaborate on the effectiveness and limitations of each module in VLSA, as well as the effectiveness and limitations of their combined use.
>
> Our motivation is to enhance model performance by minimizing information loss during the forward process of MLLMs. As outlined in Lines 46-53, we categorize the total information loss into three distinct stages: Original Information Loss (O-IL), Encoding Information Loss (E-IL), and Decoding Information Loss (D-IL). In response, we propose: (a) compressive high-resolution encoding to address O-IL without incurring additional computational costs, (b) LDM-based reconstruction to alleviate E-IL while fostering VL alignment, and (c) SSFT objectives to reduce D-IL by strengthening MLLMs' overall comprehension of visual features. Each of these three components targets a specific type of information loss and is designed to be complementary—only through the combined application of all three can we comprehensively alleviate information loss in MLLMs' forward process.. A detailed analysis of the ablation experiments in Table 4 reveals the effectiveness and limitations of each module, as well as the necessity of combining them. Below are some important conclusions we have drawn:
>
> 1. Compared to computationally expensive high-resolution methods, naively downsampling input images to low resolution degrades performance on perception-oriented benchmarks like OCR, but significantly reduces computational cost and improves performance on tasks requiring global understanding, such as high-level VQA.
> 2. Replacing naive downsampling with (a) compressive high-res encoding preserves the global understanding capability of low-resolution methods while substantially narrowing the gap with high-resolution methods in detail perception (though a gap remains), demonstrating its effectiveness in mitigating O-IL while maintaining computational efficiency.
> 3. The combination of (a) and (b) through LDM-based reconstruction further enhances detail perception, even surpassing high-resolution methods. This underscores the effectiveness of (b) in mitigating E-IL. However, this enhancement in local detail within the visual features fed into the LLM may complicate comprehension, potentially leading to a decline in global understanding (high-level VQA tasks) when compared to using (a) alone.
> 4. Combining (a), (b), and (c) SSFT objectives yields stable improvements in both detail perception and global understanding compared to high-resolution methods, demonstrating the effectiveness of (c) in mitigating D-IL and the necessity of combining all three components.
>
> **Note:** Enabling (a) compressive high-res encoding is a prerequisite for applying (b) LDM-based reconstruction and (c) SSFT objectives. When the input image size is too large, applying (b) or (c) alone would incur computational costs exceeding naive high-resolution methods (unacceptable under resource constraints), violating VLSA's design principle of computational efficiency. This is also one of the inherent limitations of components (b) and (c).
>
> Although the complete VLSA can bring consistent performance improvements to various visual-language tasks, during discussions with other reviewers, we found that its improvement on pure text unimodal tasks is limited, or even negative. The following table reports the performance of VLSA and baseline's backbone LLaMA3-8B-Instruct on pure text tasks MMLU, GSM8K, and HumanEval before and after training:
>
> | Method | MMLU (0-shot) | GSM8K (8-shot) | HumanEval (0-shot) |
> |--------|-----------|-----------|-------|
> |LLaMA3-8B|**68.4**|79.6|**62.2**|
> |LLaMA3-8B (LLaVA-Next)|66.2|77.5|60.5|
> |LLaMA3-8B (VLSA)|65.7|**79.9**|60.9|
>
> Overall, VLSA's LLM backbone demonstrates a declining trend in pure text performance after fine-tuning. Notably, the additional training objectives and model structure introduced by VLSA do not directly relate to pure text reasoning tasks, and the baseline model (LLaVA-Next) exhibits the same declining trend. Drawing on the analysis and solutions from references [1] and [2] regarding the degradation of InternVL-2's pure text performance, we believe this phenomenon is primarily due to the sub-optimal training data distribution. We anticipate that incorporating more high-quality, pure text data will help address the current performance shortcomings in pure text tasks.
>
> [1] Expanding Performance Boundaries of Open-Source Multimodal Models with Model, Data, and Test-Time Scaling
>
> [2] NVLM: Open Frontier-Class Multimodal LLMs

---

### Official Review · Reviewer_Hqtp · 2025-11-01

**Soundness:** 2
**Presentation:** 3
**Contribution:** 2
**Rating:** 4
**Confidence:** 3

**Summary:**

The VLSA (Vision-Language Semantic Alignment) framework redefines multimodal alignment by explicitly minimizing information loss throughout the visual inference process in large multimodal language models. It introduces two complementary stages: Perception Alignment, which employs a Self-Attention Perceiver (SA-Perceiver) for compressive high-resolution encoding and an LDM-based reconstructive training mechanism to preserve detailed visual semantics while maintaining efficiency; and Cognition Alignment, which enhances the LLM’s understanding of visual inputs through self-supervised fine-tuning that predicts VQ-VAE codebook indices (for high-level semantics) and pixel values (for low-level details). Integrated into models such as LLaVA, Qwen-VL, and MiniGemini, VLSA demonstrates consistent improvements across over 25 benchmarks, effectively mitigating original, encoding, and decoding information losses and improving both perception fidelity and reasoning accuracy.

**Strengths:**

- The paper clearly separates different sources of information loss across stages, making the objective of reducing each component straightforward and intuitive.
- The results show steady improvements across multiple benchmarks and model variants, indicating generally positive performance.
- The experiments demonstrate faster inference and lower computational cost, particularly on LLaVA-Next, highlighting practical efficiency gains.

**Weaknesses:**

- Although performance improvements are consistent across benchmarks, they appear incremental; gains on alternative architectures remain positive but relatively small, making it difficult to justify the added model complexity.
- The motivations behind key design choices are insufficiently explained. Despite some ablations and reasoning in the main text and appendix, the use of an LDM for reconstruction, the adoption of a VQ-VAE codebook in the alignment stage, and the single-layer MLP design for the Epigone lack clear theoretical grounding or conceptual justification. See questions for details.
- Several formulas and technical descriptions are ambiguous, and some claims—particularly those related to efficiency and attention mechanisms—are not well supported by rigorous analysis or empirical evidence. Also see the questions for details.
- The comparisons mainly focus on architectural baselines; additional baselines addressing visual–language alignment (e.g., LaViT, Ovis) should be included for a fairer evaluation.

**Questions:**

## Questions

1. Formulas
- Formula at 207 could be misleading. why use w_kv for both K and V? does that mean the weight matrices are shared, and K and V are the same?
- Formulas at 208 and 212 look non-standard. especially, what exactly is happening in 212? why is self-attention defined this way?
- For 214, how is the output split into the visual embedding and the global embedding? what determines this separation?

2. Design decisions
- I am not fully clear about the intuition behind using diffusion-based approaches for *understanding-oriented* alignment. While unified models often integrate understanding and generation jointly, most existing works rely on vision encoders (e.g., CLIP, SigLIP) for understanding and VAE–diffusion pipelines for generation. Given that MLLMs primarily focus on visual understanding rather than generation, what is the underlying motivation for introducing VAE and diffusion mechanisms in this context? What intuition supports their role in improving semantic alignment?
- For the reconstructive training, why does it need to be based on an LDM? The inclusion of a diffusion model seems to complicate both the training and generation processes. Although this is discussed in lines 250–255 and the appendix, the motivation is still unclear. Given the known training instability of diffusion models and the fact that most VLLMs focus on understanding rather than generation, this design choice feels excessive. Is there any mathematical or conceptual intuition supporting why diffusion, specifically, is suitable for this alignment objective?
- Regarding the Epigone structure, is a single-layer MLP sufficient to translate visual embeddings into prompt embeddings? Given that this mapping is highly non-trivial and involves bridging distinct semantic spaces, it is unclear whether such a simple structure can effectively perform this translation. Could the authors elaborate on this?

3. Clarifications
- The claim that “Monkey and LLaVA-UHD employ Q-former-like resamplers to reduce visual tokens, which may exacerbate E-IL” (line 133~134) needs clarification — why would reducing visual tokens through resampling necessarily increase encoding-level information loss? An explanation or supporting evidence would make this point clearer.
- Could the authors clarify the claim that “the SA-Perceiver is also responsible for achieving VL alignment” (line 215) ? Beyond being jointly trained with language-based losses, what mechanism within the SA-Perceiver itself enforces or contributes to vision–language alignment? It would be helpful to understand whether this alignment arises from the architecture’s design or solely from downstream training objectives.
- The claim that “the VQ-VAE is capable of encapsulating a broad spectrum of semantics that transcend specific, human-defined categories” (line 287-290) could be further clarified. Since VQ-VAE is a relatively early model, it is uncertain whether it can effectively capture such broad semantics compared to more recent discretization approaches. Alternative methods, such as residual quantization [1], might offer stronger representational capacity or finer semantic granularity. Also, what is the codebook size used in the experiment?


    [1] Lee et al., *Autoregressive Image Generation using Residual Quantization*, CVPR 2022.


4. Experiments
- Considering that the LLaVA family already employs multi-resolution visual processing and explicitly handles such cases, could this be the reason why the proposed method shows greater improvement on LLaVA-based models compared to other MLLMs? In other words, is the model inherently more compatible or better aligned with this architecture, leading to stronger relative gains?

5. Minor
- The citation style such as *LLaVA Liu et al. (2023a)* or *PaLI Chen et al. (2022)* feels a bit informal. Using a cleaner model-based reference format like *LLaVA (Liu et al., 2023a)* or *PaLI (Chen et al., 2022)* would look more consistent.
- Just out of curiosity, is the bottom part of Figure 1 based on an actual example? The transition between the real image and the perceived visual representation seems quite drastic.

---

> ### Author Response · Authors · 2025-11-21
>
> We extend our sincere gratitude for your constructive feedback, which is invaluable for enhancing the quality of our paper!
> ## w1:
> Regarding incremental performance improvements, this is a result of "self-constraint" stemming from our commitment to high computational efficiency. In VLSA, efficiency is primarily influenced by the resolution of the low-resolution image illustrated in Figure 2(A), which corresponds to the actual input size for the MLLM. While utilizing smaller image sizes enhances efficiency, it also increases Original Information Loss (O-IL, as defined in Line 46), which restricts further performance improvements. In our opinion, current experiments demonstrate that VLSA substantially reduces computational costs while consistently improving baselines on various tasks, which already underscores the significance of our proposed architectural and training objectives. However, if we shift focus slightly from efficiency and increase the low-resolution image size, VLSA could achieve even greater performance improvements. Following ablations on low-resolution image size will be included in the revised version:
>
> |Method|Res.|GQA|SQA-I|DocVQA|
> |:-:|:-:|:-:|:-:|:-:|
> |LLaVA-Next|Any|64.9|74.6|73.7|
> |VLSA (current)|336x336|65.3|77.5|75.2|
> |VLSA|336x672|66.8|78.1|77.8|
> |VLSA|336x1008|67.9|80.4|79.2|
> |VLSA|672x672|**69.2**|**80.8**|**80.1**|
>
> **Note:** Res. of VLSA in the table indicates the maximum size of the low-res image.
>
> Furthermore, since VLSA's pretraining introduces an auxiliary Denoising Transformer module, incorporating an additional training stage to warm up the Denoising Transformer specifically would further improve model performance compared to the current two-stage training process (we adopt standard two-stage training for fair comparison with baselines):
>
> |Method|Res.|Stages|GQA|SQA-I|DocVQA|
> |:-:|:-:|:-:|:-:|:-:|:-:|
> |LLaVA-Next|Any|2|64.9|74.6|73.7|
> |VLSA (current)|336x336|2|65.3|77.5|75.2|
> |VLSA|336x336|**3**|65.6|78.2|75.6|
> |VLSA|672x672|2|69.2|80.8|80.1|
> |VLSA|672x672|**3**|**69.5**|**81.3**|**80.2**|
>
> ## w2 & w3:
> Please refer to the responses for individual questions below.
>
> ## w4:
>
> As noted in Section 2 (Lines 147-155), LaViT reconstructs outputs from its frozen visual encoder rather than from raw images; therefore, it cannot effectively address E-IL (Encoding Information Loss) within the encoder itself. A key feature of LaViT is its use of reconstruction loss to train the Token Selector/Merger and its codebook, aiming to minimize information loss during the quantization process. Due to the structural constraints of LaViT, integrating the complete VLSA into it poses a challenge. Nonetheless, out of curiosity, we independently tested the compatibility of our Cognition Alignment with LaViT:
>
> |Method|GQA|VQA-V2|VizWiz|Nocaps|
> |:-:|:-:|:-:|:-:|:-:|
> |LaViT|$\underline{46.8}$|66.0|38.5|114.2|
> |LaViT+Cognition Alignment|$\underline{46.8}$|**66.2**|**42.3**|**118.7**|
>
> **Note:** The table shows preliminary experimental results; further tuning may enhance performance.
>
> On the other hand, Ovis improves upon previous connector (adapter)-based MLLM architectures by discretizing image tokens through a learnable vocabulary, strengthening the formal consistency between vision and text. This approach is completely orthogonal to VLSA. Following suggestion, we have supplemented experiments integrating VLSA into Ovis:
>
> |Method|MME|MMB-EN|MMMU-Val|HallusionBench|
> |:-:|:-:|:-:|:-:|:-:|
> |Ovis-Llama3-8B|2009|77.4|44.7|61.1|
> |Ovis+VLSA|**2036**|**78.4**|**45.2**|**64.2**|
>
> **Note:** Since Ovis replaces the traditional adapter with a visual vocabulary, we did not add the SA-Perceiver in these experiments and used Ovis's discretized visual embeddings as cues for reconstructive training.

---

> > ### Author Response · Authors · 2025-11-21
> >
> > ## Q1
> > (1) & (2): The reviewer's understanding is entirely correct. As illustrated in Figure 3, the SA-Perceiver employs shared K and V matrices in the cross-attention component to enhance parameter and computational efficiency. For the same reason, we omit the Q and V matrices in the self-attention component (Line 212). Although we modify the acquisition of Query, Key, and Value, our method adheres to the standard computation of attention scores. These implementation choices are backed by extensive experiments conducted during development, and we present ablation studies on variants using standard independent Q, K, and V matrices in self/cross attention in Appendix Table 8 (ex2). The findings indicate that our design improves computational efficiency without compromising performance. Intuitively, the effectiveness of omitting projection matrices may stem from the Adapter/Connector functionality of the SA-Perceiver, which does not necessitate strong learning capabilities, allowing us to reduce the number of learnable parameters without adversely affecting the outcomes.
> >
> > (3): The global embedding _P_ in Line 214 is the product of the learnable query _q_ from Line 204 after aggregating information from visual embeddings $V_{Hi}$ and $V_{Lo}$ through the SA-Perceiver. This process is analogous to obtaining the CLS token in CLIP[1]. We will describe this process more explicitly in the revised manuscript.
> >
> > [1] Learning transferable visual models from natural language supervision
> > ## Q2
> >
> > (1) & (2): There may be some misunderstanding here. As stated in Lines 100-102, our proposed Perception Alignment based on reconstructive training is not an understanding-oriented alignment method (the subsequent Cognition Alignment is the understanding-oriented alignment method). Its training objective is to enable the LDM to reconstruct the input image from random noise using the visual features input to the LLM as cues, thereby reducing information loss during image encoding (E-IL) and ensuring that the visual features comprehensively cover the visual information in the original image, thereby improving the performance upper bound of MLLMs. The additional VAE encoder and Denoising Transformer introduced by this method (which only participate in the training process and **are not utilized in model inference/testing**) are core components of LDM.
> >
> > While various autoencoder-based models, such as VQ-VAE and MAE, can also achieve similar reconstructive training objectives (as discussed in Appendix Table 10 and Lines 1011-1018), we opted to implement this with Stable Diffusion, a pretrained text-to-image LDM. The rationale behind this choice is that the primary function of the text-to-image LDM is to generate images based on textual semantics. Therefore, during reconstructive training, it will aid in aligning visual features that serve as generation (reconstruction) cues with the linguistic semantic space.
> >
> > The following addresses concerns about instability, training/testing costs, and the necessity of the LDM. First, since we use a **pretrained** LDM rather than training a randomly initialized LDM from scratch, we did not observe any instability/convergence issues during VLSA training. Second, as stated in Appendix B.3 Lines 1098-1102, we perform only a single denoising step per training iteration, and the LDM only participates in the model training process. Consequently, the additional costs brought by LDM can be substantially outweighed by the efficiencies gained through our compressive encoding. Finally, since we propose using reconstructive training to mitigate E-IL, components with image generation capabilities are indispensable. Ablations in the Appendix Table 10 demonstrate that LDM may not be strictly necessary, but at least a superior choice.
> >
> > (3): The challenge confronted by the Epigone can be analogous to the problem faced by the MLP used to bridge the vision-text space gap in adapter-based MLLMs (such as the LLaVA series and Intern-VL series). While many research studies acknowledge that a simple MLP may not fully address the demands of domain transfer, extensive community experience, including our own experiments, has demonstrated that this approach is at least effective. However, we fully agree that there is room for further improvement of the Epigone. In recent research, we have also tried more refined structures (e.g., multi-layer transformer and MoE adapters) to implement the Epigone and found some improvements. However, these complex improvement schemes introduce efficiency-performance trade-offs and exceed the scope of this paper's primary focus. In summary, we still believe that MLP is a simple and effective solution for implementing the Epigone.

---

> > > ### Author Response · Authors · 2025-11-21
> > >
> > > ## Q3
> > > (1) Q-former's resampling of visual features into fewer tokens constitutes lossy compression, as end-to-end training cannot guarantee lossless encoding. Moreover, in the early stages of training, the randomly initialized fixed-length query vectors in Q-former cannot effectively filter redundancy in visual features and retain key information, thus introducing more severe E-IL compared to MLP-based methods (which do not perform resampling, only perform mapping). These views are supported by Appendix Fig. 6 and Tab. 9(ex7). These experiments show that the Q-former converges more slowly during training and achieves significantly weaker performance compared to other projectors on the perception-oriented benchmark DocVQA. (Note that Q-former's superior performance on some global understanding tasks only indicates better global information aggregation capabilities and does not contradict the conclusion that it produces more severe detail information loss during encoding.)
> > >
> > > (2) The SA-Perceiver assumes the same Adapter/Connector role in the MLLM architecture as the MLP projector and Q-former. These modules are responsible for achieving VL alignment, but have no perception/processing capabilities for textual features and only operate on visual features. Therefore, their VL alignment capabilities arise solely from end-to-end training objectives. The structural design of the SA-Perceiver primarily serves to efficiently condense the information from high-resolution images into their downsampled counterparts.
> > >
> > > (3) It is important to clarify that the comparison referenced in Lines 287-290 pertains to recently proposed auxiliary training objectives for MLLMs (rather than specific vision encoders). Previous objectives, such as visual grounding and segmentation (Lines 91-94), primarily focus on specific attributes of images. In contrast, the codebook indices of input images provided by the pretrained VQ-VAE can be used to restore the original images, thus naturally helping MLLMs understand visual semantics more comprehensively. This represents a structural advantage inherent to the task definition, rather than an improvement resulting from the use of particular powerful models. Furthermore, our method is not limited to VQ-VAE models; any quantization encoder that can provide discrete labels can be applied to Cognition Alignment. As the reviewer mentioned, more powerful quantization encoders may bring stronger performance improvements. Nevertheless, this paper focuses primarily on demonstrating the effectiveness of this novel pretraining objective. However, we commit to exploring better model choices in depth in the future to serve the research community better.
> > >
> > > The codebook size of the pretrained VQ-VAE used in experiments is 4096.
> > >
> > > ## Q4:
> > > Several factors may contribute to this phenomenon: (1) The hyperparameters utilized in the current experiments are specifically tailored for LLaVA. (2) Different MLLMs train on sets that vary in both the quantity and diversity of images. However, our Cognition Alignment requires the model to learn to predict the codebook indices of images, so the benefits of this objective will depend on the scale and distribution of the images used. (3) VLSA enhances model performance by minimizing information loss. Different models experience varying degrees of information loss, which affects the magnitude of the gains achieved through VLSA. Nevertheless, the current experiments demonstrate that VLSA consistently yields positive gains across various MLLM architectures, underscoring the effectiveness and value of this approach.
> > >
> > > ## M1:
> > > We thank the reviewer for the suggestion. This has been corrected in the revised manuscript.
> > >
> > > ## M2:
> > > Yes, Figure 1 shows a real example, but it is a carefully selected worst-case example.

---

### Official Review · Reviewer_2bhA · 2025-11-01

**Soundness:** 3
**Presentation:** 4
**Contribution:** 2
**Rating:** 6
**Confidence:** 4

**Summary:**

This work proposes two additional alignment stage to improve the vision-language alignment between the visual encoder and language decoder in common LVLMs. First of these stages, "Perception Alignment", aims to reduce the potential losses occurring during the visual encoding process through aligning the initial visual features with encoder output features whereas the second of these stages, "Cognition Alignment", aims to mitigate any such losses occurring during the decoding phase. The work provides several quantitative and qualitative results to support its claims.

**Strengths:**

Here are the main strengths of this work:

- The central methodology of the paper, combining the three existing ideas of low-res/hig-res alignment, reconstructive training and using auxiliary self-supervised objectives for the decoder is novel and interesting.
- The formulation of the modality alignment problem in three stages in this manner, from the initial lossy compression errors to decoding phase errors is clear and technically sound.
- The narrative of the work is easy to follow, notations are clear and technical details are discussed extensively.
- Several results demonstrate the effectiveness of the proposed method, e.g. some of the results on Tables 1 and 2.

**Weaknesses:**

Here are the main weaknesses of the work:

**W1: Complex Pipeline with Potentially Redundant Aspects:** Although the overall pipeline is novel and interesting, the work introduces several hard-to-engineer phases, with performance gains failing to justify them fully. While some parts of this pipeline is well-known to be working well in the field (e.g. the low-res/high-res setting, also evidenced by the ablation results on Table 4), several parts seem to bring relatively marginal improvements (same ablation table highlights this to an extent). With the additional parameters, hyperparameters and other training settings introduced by the method, this multi-stage pipeline could be undesirable for downstream applications.

**W2: Lack of Empirical Validation:** The work lacks empirical validation in several dimensions:

- The results presented in the work are still text-heavy benchmarks, as discussed by [A] in the literature for example. Given the goal of the work around improving the vision-language alignment, having more visual-heavy benchmarks would have been better, e.g. OCR-VQA or RefCOCO.

- Furthermore, the improvements brought in by this multi-stage pipeline are relatively low, except for a few others, such as VizWizVQA. Having confidence intervals around the evals with marginal improvements (e.g. GQA) would have been great.

- Evidence for “information-loss reduction” is indirect. Gains on VQA-like tasks don’t isolate O/E/D-IL. There’s no direct measure (e.g., reconstruct-from-embedding PSNR/LPIPS) showing retained information, making the exact contributions of the multiple stages hard to grasp.

- Finally, reporting the vision-only performance (e.g. image recognition of the visual encoder) and the text-only performance of the individual components would also have been great to see the effects of the additional alignment stages introduced on them.


---
[A] Tong, P., Brown, E., Wu, P., Woo, S., IYER, A. J. V., Akula, S. C., ... & Xie, S. (2024). Cambrian-1: A fully open, vision-centric exploration of multimodal llms. Advances in Neural Information Processing Systems, 37, 87310-87356.

**Questions:**

- How well do you think VLSA work for OCR-heavy benchmarks, e.g. OCR-VQA, and benchmarks requiring finer-grained visual analysis compared to the ones presented in the work, e.g. RefCOCO?

- How well do you think that the end visual encoder you get is performing on its own, e.g. in its image recognition capabilities?

- Orthogonally to this question, Cambrian [A] also introduced a very neat methodology for checking, involving adapting the model with a frozen LLM. How well do you think the visual encoder and the projector in this work would perform under Cambrian's settings?

- Does VLSA bring in a regression in text-only benchmarks, e.g. on HellaSwag?

---
[A] Tong, P., Brown, E., Wu, P., Woo, S., IYER, A. J. V., Akula, S. C., ... & Xie, S. (2024). Cambrian-1: A fully open, vision-centric exploration of multimodal llms. Advances in Neural Information Processing Systems, 37, 87310-87356.

---

> ### Author Response · Authors · 2025-11-21
>
> We appreciate the reviewer's acknowledgment of our work. We believe the following responses will further enhance the reviewer's confidence in our paper.
>
> ## w1:
> We will address the reviewer's concerns from two perspectives.
>
> ### Complexity of VLSA and Feasibility in Downstream Applications:
>
> Proposing a simple and easy-to-use performance enhancement plugin constitutes one of the primary motivations underpinning our research. First, it is important to clarify that within our proposed two-stage alignment framework, VLSA for mitigating information loss, the term "stage" does not denote actual training phases but rather alludes to the distinct alignment objectives and their respective aims. Specifically, Perception Alignment aligns visual embeddings with visual inputs, aiming to reduce Original Information Loss (O-IL) and Encoding Information Loss (E-IL). On the other hand, Cognition Alignment aligns the LLM's reasoning process with visual embeddings, aiming to reduce Decoding Information Loss (D-IL). These two alignments have no strict sequential relationship and **do not introduce additional training stages**. Practically, our approach emphasizes strict adherence to the training configurations of the baseline model, which includes hyperparameters, training stages, and data utilization. Consequently, Perception Alignment is activated during the training of the visual encoder or Adapter, while Cognition Alignment is employed when updating the parameters of the LLM. Taking the LLaVA series as an example, it has two training stages: the first stage trains only the Adapter, and the second stage trains all model parameters. Therefore, we enable only Perception Alignment in the first stage and enable both Perception and Cognition Alignment in the second stage.
>
> Second, regarding additional parameters, VLSA maintains relative parameter efficiency: (1) As detailed in Appendix B.3 (Lines 1087-1097), SA-Perceiver has fewer parameters and higher computational efficiency compared to common Adapters like MLP or Q-former. (2) The VQ-VAE used in Cognition Alignment only participates in offline data annotation and is not an actual model component. (3) The VAE and Denoising Transformer introduced by Reconstructive training only participate in the model training stage and **incur no overhead during testing**. Moreover, the VAE remains frozen throughout, allowing offline image encoding in resource-constrained scenarios.
>
> Finally, regarding additional hyperparameters, **we did not introduce any additional hyperparameters** in the paper. While there does exist an important variable affecting performance—the resolution of the low-res image illustrated in Figure 2(A)—we found that the current naive default setting achieves optimal efficiency gains and stable performance improvements across various baselines. This means VLSA can obtain stable gains without task-specific or architecture-specific tuning, making it convenient for use in various scenarios. However, it still leaves flexibility for real-world applications, such as relaxing efficiency constraints by increasing the low-res image size to boost performance further. In subsequent responses, we also provide supplementary ablation experiments on this variable.
>
> ### Performance Gains and Necessity of Each Component:
>
> In this paper, we propose: (a) compressive high-resolution encoding to address O-IL without incurring additional computational costs, (b) LDM-based reconstruction to alleviate E-IL while fostering VL alignment, and (c) SSFT objectives to reduce D-IL by strengthening MLLMs' overall comprehension of visual features. Each of these three components targets a specific type of information loss and is designed to be complementary—only through the combined application of all three can we comprehensively alleviate information loss in MLLMs' forward process.
>
> Furthermore, our experiments reveal that **using any single component alone is a double-edged sword** (i.e., significantly improving certain capabilities while harming others). Only by applying all three components together can we achieve comprehensive improvements across various downstream task types. In Table 4 and Section 4.2, we present comprehensive ablations illustrating the individual effectiveness, limitations, and the importance of their integration. To save the reviewer's time, we have summarized some key arguments and conclusions below:

---

> > ### Author Response · Authors · 2025-11-21
> >
> > 1. Compared to computationally expensive high-resolution methods, naively downsampling input images to low resolution degrades performance on perception-oriented benchmarks like OCR, but significantly reduces computational cost and improves performance on tasks requiring global understanding, such as high-level VQA.
> > 2. Replacing naive downsampling with (a) compressive high-res encoding preserves the global understanding capability of low-resolution methods while substantially narrowing the gap with high-resolution methods in detail perception (though a gap remains), demonstrating its effectiveness in mitigating O-IL while maintaining computational efficiency.
> > 3. The combination of (a) and (b) through LDM-based reconstruction further enhances detail perception, even surpassing high-resolution methods. This underscores the effectiveness of (b) in mitigating E-IL (as it impacts only the visual encoder without altering the input image size or the LLM). However, this enhancement in local detail within the visual features fed into the LLM may complicate comprehension, potentially leading to a decline in global understanding (high-level VQA tasks) when compared to using (a) alone.
> > 4. Combining (a), (b), and (c) SSFT objectives yields stable improvements in both detail perception and global understanding compared to high-resolution methods, demonstrating the effectiveness of (c) in mitigating D-IL and the necessity of combining all three components.
> >
> > **Note:** Enabling (a) compressive high-res encoding is a prerequisite for applying (b) LDM-based reconstruction and (c) SSFT objectives. When the input image size is too large, applying (b) or (c) alone would incur computational costs exceeding naive high-resolution methods (unacceptable under resource constraints), violating VLSA's design principle of computational efficiency.
> >
> > Finally, regarding relatively marginal improvements, this is a result of "self-constraint" stemming from our commitment to high computational efficiency. In VLSA, efficiency is primarily influenced by the resolution of the low-resolution image illustrated in Figure 2(A), which corresponds to the actual input size for the MLLM. While utilizing smaller image sizes enhances efficiency, it also increases O-IL, which restricts the upper performance bound of MLLMs. In our opinion, current experiments demonstrate that VLSA substantially reduces computational costs while consistently improving baselines on various tasks, which already underscores the significance of our proposed architectural and training objectives. However, if we shift focus slightly from efficiency and increase the low-resolution image size, VLSA could achieve even greater performance improvements. Following ablations on low-resolution image size will be included in the revised version:
> >
> > |Method|Res.|GQA|SQA-I|DocVQA|
> > |:-:|:-:|:-:|:-:|:-:|
> > |LLaVA-Next|Any|64.9|74.6|73.7|
> > |VLSA (current)|336x336|65.3|77.5|75.2|
> > |VLSA|336x672|66.8|78.1|77.8|
> > |VLSA|336x1008|67.9|80.4|79.2|
> > |VLSA|672x672|**69.2**|**80.8**|**80.1**|
> >
> > **Note:** Res. of VLSA in the table indicates the maximum size of the low-res image.

---

> ### Author Response · Authors · 2025-11-21
>
> ## w2：
> ### (1):
> In this paper, we have evaluated VLSA on visual-heavy benchmarks including DocVQA (document understanding, Tab.4), ChartQA (chart understanding, Tab.4), TextVQA (scene text recognition, Tab.1), and AI2D (scientific diagram understanding, Tab.4). The experimental results demonstrate that VLSA significantly improves the model's fine-grained perception capabilities while maintaining computational efficiency comparable to low-resolution settings, even achieving stable improvements over the standard high-resolution methods used in baselines (while reducing FLOPs by 77% and system latency by 50%). Furthermore, as stated in Response w1, VLSA's performance would improve further if efficiency constraints are relaxed.
>
> In response to the reviewer's suggestions, we have also included performance comparisons for InfoVQA (infographic understanding) and OCRBench (OCR capabilities). The datasets RefCOCO and OCR-VQA mentioned by the reviewer were not evaluated, as they are already part of the training set.
>
> |Method|Res.|InfoVQA|OCRBench|
> |:-:|:-:|:-:|:-:|
> |LLaVA-Next|336x336|40.6|419|
> |LLaVA-Next|Any|44.1|510|
> |VLSA|336x336|44.6|554|
> |VLSA|672x672|**47.2**|**563**|
>
> ### (2):
> Following the suggestion, we report VLSA's average performance and variance across three runs (seeds set to 42, 456, 2024) on GQA, VQA-V2, and POPE:
> |Method|Res.|GQA|VQA-V2|POPE|
> |:-:|:-:|:-:|:-:|:-:|
> |LLaVA-Next|336x336|62.3|77.1|85.4|
> |LLaVA-Next|Any|64.9|82.4|87.7|
> |VLSA|336x336|65.37±0.21|83.47±0.15|88.76±0.40|
> |VLSA|672x672|69.2|85.1|89.5|
>
> Note: Due to time constraints, results under the new setting (672x672) are not statistical values from multiple runs.
>
> ### (3):
> Completely isolating O/E/D-IL for analysis is indeed very difficult. Our paper reports qualitative evidence in Figure 4 that VLSA can significantly mitigate E-IL. As a supplement, we calculated the reconstruct-from-embedding PSNR for VLSA and baseline models. Specifically, we randomly selected 1,000 images from the training set and computed the average PSNR of their corresponding reconstructed images:
>
> |Method|Res.|PSNR|
> |:-:|:-:|:-:|
> |LLaVA-Next|Any|9.02|
> |VLSA|336x336|**20.18**|
>
> ### (4):
> We thank the reviewer for this suggestion. We evaluate on ImageNet-1K zero-shot classification to directly assess visual encoding quality:
>
> | Method | Metric| ImageNet-1K | Notes |
> |--------|-----------|-----------|-------|
> | LLaVA-Next |Open-World Setting| 31.3 | Full model |
> | VLSA |Open-World Setting | **36.0** | Full model |
>
> | Method | Metric| ImageNet-1K | Notes |
> |--------|-----------|-----------|-------|
> | CLIP ViT-L/14 |Closed-World Setting| **74.8** |Vision encoder for LLaVA & VLSA before finetuning |
> | LLaVA-Next |Closed-World Setting| 47.4 | Vision encoder only |
> | VLSA |Closed-World Setting | 53.2 | Vision encoder only |
> | VLSA |Closed-World Setting | 63.8 | Vision encoder only (Cambrian's settings) |
>
> Note: the Open-world setting means the label set is not provided and the Closed-world setting means classes are concatenated in the prompt.
>
> The experimental results demonstrate that VLSA, by minimizing visual information loss, significantly enhances image recognition capabilities compared to baseline models. It is essential to note that while we observe a decrease in the CLIP encoder's image recognition performance after VLSA's training—although it still outperforms baseline models—we consider this decline to be anticipated. During VLSA's training, the optimization objective for the CLIP vision encoder shifts from the initial global contrastive learning to a reconstructive training approach that emphasizes detailed semantic understanding of images (in addition to incorporating language modeling loss). However, the original optimization objective has a closer alignment with the test task. Moreover, reference [1] highlights a similar phenomenon and provides an explanation:
>
>  "The primary cause is data-related: critical information for image classification is encoded in the VLM's latent space but can only be effectively decoded with sufficient training data."
>
> This suggests that vision encoders within MLLMs require additional fine-tuning to better adapt to image recognition tasks, as the current zero-shot setting fails to fully harness their potential.

---

> > ### Author Response · Authors · 2025-11-21
> >
> > The following table reports the performance of VLSA and baseline's backbone LLaMA3-8B-Instruct on pure text tasks MMLU, GSM8K, and HumanEval before and after training:
> >
> > | Method | MMLU (0-shot) | GSM8K (8-shot) | HumanEval (0-shot) |
> > |--------|-----------|-----------|-------|
> > |LLaMA3-8B|**68.4**|79.6|**62.2**|
> > |LLaMA3-8B (LLaVA-Next)|66.2|77.5|60.5|
> > |LLaMA3-8B (VLSA)|65.7|**79.9**|60.9|
> >
> > Overall, VLSA's LLM backbone demonstrates a declining trend in pure text performance after fine-tuning. Notably, the additional training objectives and model structure introduced by VLSA do not directly relate to pure text reasoning tasks, and the baseline model exhibits the same declining trend. Drawing on the analysis and solutions from references [2] and [3] regarding the degradation of InternVL-2's pure text performance, we believe this phenomenon is primarily due to the sub-optimal training data distribution. We anticipate that incorporating more high-quality, pure text data will help address the current performance shortcomings in pure text tasks.
> >
> > [1] Why are Visually-Grounded Language Models Bad at Image Classification?
> >
> > [2] Expanding Performance Boundaries of Open-Source Multimodal Models with Model, Data, and Test-Time Scaling
> >
> > [3] NVLM: Open Frontier-Class Multimodal LLMs
> >
> > ## Q1 & Q2 & Q4:
> > Please refer to the preceding responses addressing the paper's weaknesses.
> >
> > ## Q3:
> > In the supplementary results of Response w2, we assessed the performance of the vision encoder trained under Cambrian's settings (i.e.,  with a frozen LLM). Specifically, we included all available training data (1.3 M) in the first stage of VLSA’s training, allowing for Perception Alignment while excluding Cognition Alignment to train the model. The experiments demonstrated that this approach further enhances the performance of the encoder, although it still does not surpass the performance of the unfine-tuned CLIP model. This improvement is anticipated, as targeted parameter updates for the vision encoder tend to yield greater benefits than end-to-end training.

---

### Author Response · Authors · 2025-12-04
**Summary of Reviewer Feedbacks**

We would like to express our sincere gratitude to all reviewers and ACs for their dedicated efforts throughout these hard times. Your valuable feedback has significantly enhanced the quality of our manuscript.

In this paper, we propose the VLSA framework, which serves as an easy-to-use plugin for existing MLLMs, not only delivering consistent performance/efficiency gains across diverse model architectures but also offering novel insights and practical solutions for advancing modality alignment by reducing information loss during the inference process.

During the rebuttal phase, we conducted essential additional experiments and provided clear clarifications. We are confident that these responses can adequately address the remaining concerns.

We sincerely hope this work can further contribute to the ongoing progress of the MLLM research community.

# Key strengths highlighted by reviewers

1. **Novel methodology with clear technical framework**
\
Reviewers appreciated the central methodology, which combines compressive high-resolution encoding, reconstructive training, and novel auxiliary self-supervised objectives. The formulation of the modality alignment problem in two stages, from initial lossy encoding errors to decoding phase errors, is clear and technically sound.
\
_(Reviewers 2bhA, HqtP, 1pvz, sE3s)_

2. **Clear decomposition of information loss**
\
The paper clearly separates different sources of information loss (O-IL, E-IL, D-IL) across stages, making the objective of reducing each component straightforward and intuitive. This decomposition motivated the two-stage alignment (three-component) design with concrete framework construction and training objectives.
\
_(Reviewers HqtP, 1pvz, sE3s)_

3. **Practical efficiency and lightweight design**
\
VLSA maintains high computational efficiency with a constant 576-token visual sequence and deactivated auxiliary components at test time. The experiments demonstrate faster inference and lower computational cost, particularly reducing 77% FLOPs and 50% system latency compared to high-resolution baselines.
\
_(Reviewers HqtP, 1pvz, sE3s)_

4. **Consistent empirical improvements**
\
Reviewers noted consistent improvements across 25 benchmarks (GQA, VQA-V2, DocVQA, ChartQA, etc.) and 7 MLLM architectures (LLaVA, InternVL, Qwen2.5-VL) when integrating VLSA, demonstrating its robustness and broad applicability.
\
_(Reviewers 2bhA, HqtP, 1pvz, sE3s)_

5. **Clear presentation and comprehensive analyses**
\
The narrative is easy to follow with clear notations and extensive technical details. The paper includes comprehensive ablations on proposed components (illustrating the individual effectiveness, limitations, and the importance of their integration), as well as layer configurations, and multiple encoder variants that support the method design.
\
_(Reviewers 2bhA, HqtP)_

---

> ### Author Response · Authors · 2025-12-04
>
> # Main questions and our responses
>
> 1. **Complexity and necessity of multi-component design**
> \
> Reviewers questioned whether the multi-stage pipeline introduced too much complexity with marginal improvements, and whether all three components (compressive encoding, LDM reconstruction, SSFT objectives) were necessary together. We clarified that "stage" refers to alignment objectives rather than training phases, and VLSA does not introduce additional training stages beyond the baseline. Each component targets a specific type of information loss (O-IL, E-IL, D-IL) and is complementary. Ablations in Table 4 demonstrate that using any single component alone is a double-edged sword - improving certain capabilities while harming others. Only combining all three achieves comprehensive improvements across both detail perception and global understanding tasks. The apparently marginal improvements stem from our commitment to high computational efficiency; new results reported during the rebuttal demonstrate that relaxing efficiency constraints by increasing low-res image size yields substantially larger performance gains.
> \
> _(Reviewers 2bhA, HqtP, sE3s)_
>
> 2. **Design rationale for LDM-based reconstruction**
> \
> Reviewers asked why diffusion models were necessary for understanding-oriented alignment, noting concerns about training instability and complexity. We explained that Perception Alignment via reconstructive training aims to reduce E-IL by enabling the LDM to reconstruct input images from visual features, ensuring visual features comprehensively cover original image information. While alternatives like VQ-VAE or MAE can achieve similar objectives (Appendix Table 10), we chose pretrained Stable Diffusion because text-to-image LDMs help align visual features toward linguistic semantic space during reconstruction. Using a pretrained (not randomly initialized) LDM avoids instability, and performing only a single denoising step per training iteration keeps costs manageable. The LDM components only participate in training and incur no overhead during inference.
> \
> _(Reviewers HqtP)_
>
> 3. **Empirical validation on visual-heavy benchmarks and more evaluation metrics**
> \
> Reviewers requested evaluation on more visually heavy benchmarks, provide confidence intervals on benchmarks with marginal improvements, and the PSNR of reconstructed images. Besides DocVQA, ChartQA, TextVQA, and AI2D, which are included in the manuscript, we added results on InfoVQA (infographic understanding) and OCRBench (OCR capabilities), showing VLSA achieves substantial improvements (e.g., InfoVQA: 44.1->47.2; OCRBench: 510->563). We also reported confidence intervals across three runs (seeds 42, 456, 2024) on GQA, VQA-V2, and POPE, demonstrating stable gains with small variance. Additionally, we provided reconstruct-from-embedding PSNR (baseline: 9.02, VLSA: 20.18) as direct evidence of E-IL reduction.
> \
> _(Reviewer 2bhA)_
>
> 4. **Vision encoder and text-only performance**
> \
> Reviewers inquired about the quality of the vision encoder after VLSA training and whether text-only performance had regressed. We evaluated the vision encoder on ImageNet-1K zero-shot classification, showing VLSA improves both full-model (31.3->36.0 open-world) and vision-encoder-only performance (47.4->53.2 closed-world). Under Cambrian's frozen-LLM setting, the vision encoder achieves even higher performance. On pure-text benchmarks (MMLU, GSM8K, HumanEval), VLSA exhibits a slight degradation, similar to the baseline LLaVA-Next. Following analysis of the InternVL-2 literature, this is primarily due to the distribution of training data rather than specific issues with VLSA, and can be addressed by adding more high-quality text data.
> \
> _(Reviewer 2bhA)_
>
> 5. **Technical clarifications on formulas and design choices**
> \
> Reviewers requested clarifications on formula notation (shared K/V matrices in Line 207, self-attention definition in Line 212) and design choices (single-layer MLP for Epigone, VQ-VAE codebook size). We confirmed that SA-Perceiver uses shared K/V weight matrices and omits Q/V matrices in self-attention for parameter efficiency, supported by ablations in Appendix Table 8 (ex2). For the Epigone, we acknowledged that while a single-layer MLP may not be optimal, it is proven effective in practice (similar to MLPs in LLaVA/InternVL adapters), and more complex structures introduce efficiency-performance trade-offs beyond this paper's scope. The VQ-VAE codebook size used is 4096.
> \
> _(Reviewers HqtP, sE3s)_

---

> ### Author Response · Authors · 2025-12-04
>
> 6. **Integration with other VL alignment methods**
> \
> Reviewers suggested comparing with other alignment-focused baselines, such as LaViT and Ovis. We clarified that the comparisons have been made in Sec 2, and added experiments showing our Cognition Alignment can further improve LaViT (e.g., VizWiz: 38.5->42.3, Nocaps: 114.2->118.7) and full VLSA integration with Ovis yields consistent gains (MME: 2009->2036, HallusionBench: 61.1->64.2), demonstrating VLSA's compatibility with these orthogonal architectural innovations.
> \
> _(Reviewer HqtP)_
>
> 7. **Handling conflicts between local details and global semantics**
> \
> Reviewers questioned how SA-Perceiver prevents high-res local details from overwhelming low-res global semantics. We explained that SA-Perceiver uses Low-Res Embeddings as Queries to aggregate information from High-Res Embeddings via cross-attention, organizing local details within the structure of global semantics. This design mitigates the challenge of modeling detail-rich visual content solely through LLM causal attention and can be regarded as an inductive bias that enhances the ability to handle local details. Besides, our SSFT objectives further help the LLM comprehend both types of information, and end-to-end training automatically balances their importance. Ablations (Figure 6, Table 9) show this approach yields better convergence and performance than simple concatenation methods.
> \
> _(Reviewer sE3s)_

---

### Meta-Review · Area_Chair_LZbN · 2026-01-10

**Summary:**

Reviewers express positive attitude on the technical novelties.

Some concerns are also raised by the reviewers.
1. The pipeline is complex and contains hard-to-engineer phases [Reviewer 2bhA], the contributions is small and hard to justify the added model complexity [Reviewer Hqtp],
2. The evaluations are not sufficient. Missing benchmarks, detailed ablation studies, [Reviewer 2bhA, qpvz, sE3s], missing more baselines [Reviewer Hqtp],
3. Motivations are not clearly addressed [Reviewer Hqtp, sE3s],
4. Technical novelty. [Reviewer sE3s].


I read the reviewers' comments and the authors' rebuttal information. I think that the technical novelties are not significant. And the added some components make the whole pipeline complicated with each component contribution incremental, although some what effective yet technically incremental.

**Reviewer Concerns:**

The reviewers comments on insufficient evaluations can be well addressed, such as missing benchmarks, missing baselines, missing detailed ablation studies.

Motivations-method connection can be addressed.

The technical novelty and the complexity of the pipeline, as well the incremental contributions and evaluations are still outstanding.

**Reviewer Scores:**

The reviewers may stand on their previous rating, as the two reviewers expressing negative ideas, mainly concerns on the technical novelties, technical pipelines.

---

### Decision · Program_Chairs · 2026-01-26

Reject